# Mutation saturation for fitness effects at human CpG sites

**Ipsita Agarwal[1]\*, Molly Przeworski[1,2]\***

[1]Department of Biological Sciences, Columbia University, New York, United States;
[2]Department of Systems Biology, Columbia University, New York, United States

**Abstract** Whole exome sequences have now been collected for millions of humans, with the related goals of identifying pathogenic mutations in patients and establishing reference repositories of data from unaffected individuals. As a result, we are approaching an important limit, in which datasets are large enough that, in the absence of natural selection, every highly mutable site will have experienced at least one mutation in the genealogical history of the sample. Here, we focus on CpG sites that are methylated in the germline and experience mutations to T at an elevated rate of $\sim 10^{-7}$ per site per generation; considering synonymous mutations in a sample of 390,000 individuals, $\sim 99\,\%$ of such CpG sites harbor a C/T polymorphism. Methylated CpG sites provide a natural mutation saturation experiment for fitness effects: as we show, at current sample sizes, not seeing a non-synonymous polymorphism is indicative of strong selection against that mutation. We rely on this idea in order to directly identify a subset of CpG transitions that are likely to be highly deleterious, including ~27 % of possible loss-of-function mutations, and up to 20 % of possible missense mutations, depending on the type of functional site in which they occur. Unlike methylated CpGs, most mutation types, with rates on the order of $10^{-8}$ or $10^{-9}$, remain very far from saturation. We discuss what these findings imply for interpreting the potential clinical relevance of mutations from their presence or absence in reference databases and for inferences about the fitness effects of new mutations.

**\*For correspondence:**
ia2337@columbia.edu (IA);
mp3284@columbia.edu (MP)

## Introduction

A central goal of human genetics is to identify pathogenic mutations and predict how likely they are to cause disease. To this end, exome sequencing in cases and controls is often used to help identify variants with potentially large effects on disease risk (*Rauch et al., 2012*; *Sanders et al., 2012*; *Need et al., 2012*; *Akbari et al., 2021*). Even where this approach yields an enrichment of variants in cases, however, the specific subset of mutations that contributes to disease often remains unknown; similarly, in individual patients, sequencing habitually yields candidate mutations of which the significance is unclear (*Richards et al., 2015*; *Harrison et al., 2021*).

Numerous scores have therefore been developed to help prioritize among candidate mutations, based on protein structure, functional annotations, evolutionary patterns, or other features (*Cooper et al., 2005*; *Adzhubei et al., 2010*; *Pollard et al., 2010*; *McLaren et al., 2016*; *Ioannidis et al., 2016*; *Rentzsch et al., 2019*). In particular, a common approach to pinpoint sites at which mutations are likely to be pathogenic is to examine whether they appear to be under purifying selection. For instance, comparisons of sequences across species have been widely used to identify highly conserved genomic regions maintained by selection over millions of years, presumably because of their functional importance (*Cooper et al., 2005*; *Pollard et al., 2010*; *Boffelli et al., 2003*; *Siepel et al., 2005*).

The same general approach is also useful when applied within humans, where information about purifying selection is contained in whether or not a site is segregating a mutation and at what frequency (*Sawyer and Hartl, 1992*; *Eyre-Walker and Keightley, 2007*; *Boyko et al., 2008*; *Williamson et al.,*

*2005*; *Lek et al., 2016*; *Karczewski et al., 2020*; *Yi et al., 2010*). For this application, however, the low diversity levels in the genome pose a major difficulty, as a site may be monomorphic simply by chance, that is when mutations at that site have no fitness consequences at all or, at the other extreme, because the mutations are embryonically lethal. In particular, because most sites are monomorphic in samples of hundreds or even thousands of humans, there is little information to distinguish sites under strong selection from those at which mutations are only weakly deleterious.

With a view to capturing natural variation at a larger number of sites in the genome and identifying more mutations with large effects on disease risk, there have been extensive efforts to collate available exome sequences from hundreds of thousands of individuals (*Lek et al., 2016*; *Karczewski et al., 2020*; *Dewey et al., 2016*; *Szustakowski, 2020*; *Van Hout et al., 2020*; *Taliun et al., 2021*). These efforts were also motivated by the idea that public repositories composed of relatively healthy adults not ascertained for a specific severe disease can serve as reference datasets, such that seeing a variant of unknown function in these datasets is indicative of it being benign (*Lek et al., 2016*; *Claussnitzer et al., 2020*; *Ghouse et al., 2018*). The validity of that assumption remains to be evaluated, however, especially as the repositories grow in size.

Beyond their utility in human genetics, these datasets provide an unprecedented opportunity to learn about the fitness effects of new mutations. Modeling the distribution of fitness effects (DFE) has a long history in population genetics (*Sawyer and Hartl, 1992*; *Eyre-Walker and Keightley, 2007*; *Otto, 2000*), but until recently, inferences were based on genetic variation in samples of at most a couple of thousand chromosomes (*Eyre-Walker and Keightley, 2007*; *Boyko et al., 2008*; *Williamson et al., 2005*; *Eyre-Walker et al., 2006*; *Kim et al., 2017*). As is well appreciated, the fitness effects at the few sites segregating at such sample sizes are a small and biased draw from the DFE and thus the inferred distribution of fitness effects is unlikely to recapitulate the true DFE in the genome. Moreover, for lack of sufficient information with which to distinguish weakly from strongly selected mutations, a number of approaches have relied on a specific and arbitrary parametric form for the distribution of fitness effects across sites. In that regard, not only do inferences based on small samples result in relatively noisy parameter estimates, the results can be misleading, especially about the fraction of sites under strong selection (*Kim et al., 2017*). Current samples in humans may allow for these limitations to start to be overcome.

Motivated by these considerations, we focus on a class of mutations known to experience mutations an order of magnitude more frequently than other types of sites in the human genome: CpG sites that are methylated in the germline (*Duncan and Miller, 1980*; *Nachman and Crowell, 2000*; *Kong et al., 2012*; *Figure 1—figure supplement 1*). We use these sites as a test case for what can be learned about selection when neutral sites are saturated, i.e., have all experienced at least one mutation in the history of the sample, and draw out implications for the interpretation of mutations as pathogenic and for inferences about fitness effects.

## Results
### Mutation saturation at CpGs

An attractive feature of methylated CpG (mCpG) sites is that a single mechanism, the spontaneous deamination of methyl-cytosine, is believed to underlie the uniquely high rate of C > T mutations at these sites (*Duncan and Miller, 1980*); thus, germline methylation at CpG sites is strongly predictive of their mutability (*Kong et al., 2012*; *Jónsson et al., 2017*; *Gao et al., 2019*; *Figure 1—figure supplement 2*). Here, we define 'methylated' CpG sites in exons as those that are methylated ≥65 % of the time in both testes and ovaries. For these ~1.1 million sites (of 1.8 million total CpG sites in sequenced exons), we calculate a mean haploid, autosomal C > T mutation rate of $1.17 \times 10^{-7}$ per generation using de novo mutations (DNMs) in a sample of ~2900 sequenced parent-offspring trios (Materials and methods, *Figure 1—figure supplements 1–2*, *Halldorsson et al., 2019*).

Although methylation levels are the dominant predictor of mutation rates at CpG sites, they are not the only influence. Notably, CpG transitions differ somewhat in their mutation rates based on their trinucleotide context (*Figure 1—figure supplement 3a*; *Aggarwala and Voight, 2016*); even so, they are consistently an order of magnitude higher than the genome average (*Kong et al., 2012*). Broader scale features, such as replication timing, have also been reported to shape mutation rates (*Stamatoyannopoulos et al., 2009*; *Smith et al., 2018*). Nonetheless, considering methylated CpGs

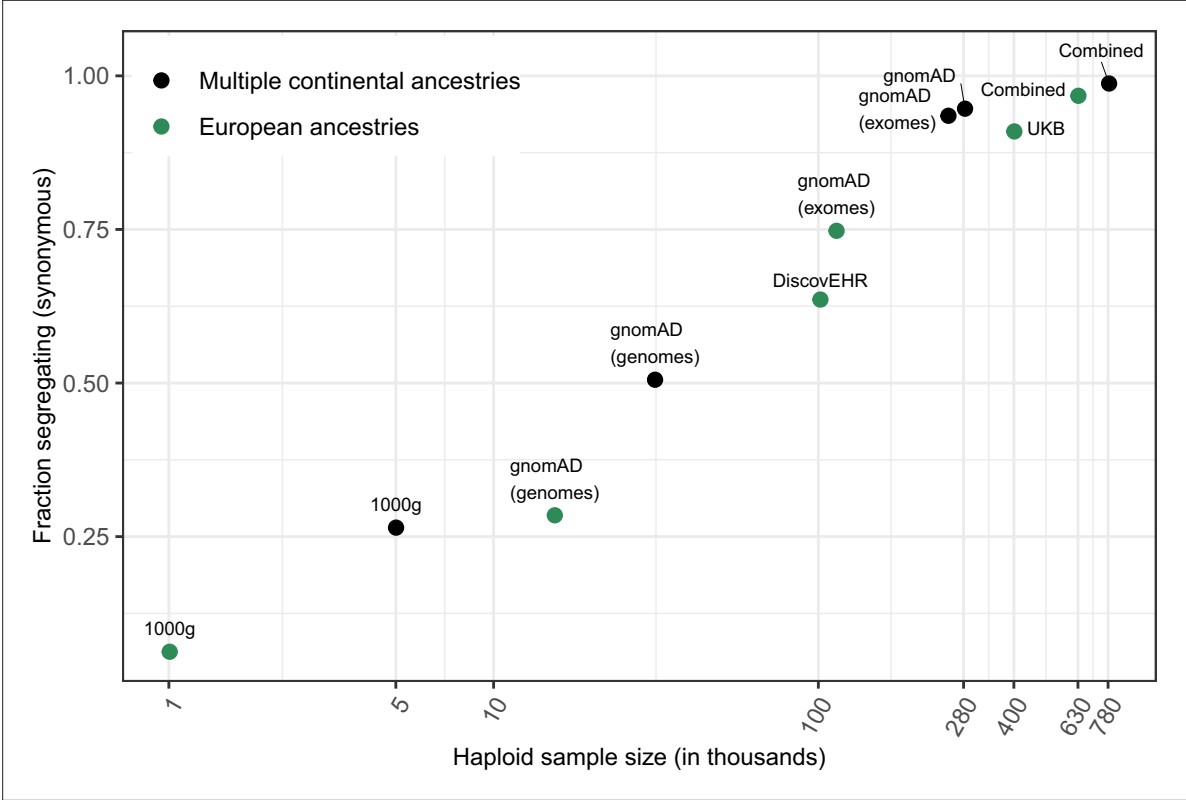

**Figure 1.** Fraction of methylated CpG sites that are polymorphic for a transition, by sample size. The combined dataset encompasses three non-overlapping data sources: gnomAD (v2.1), the UK Biobank (UKB), and the DiscovEHR cohort. 'European' samples include the populations designated as 'EUR' in 1000 Genomes, 'Non-Finnish European' subsets of exome and whole genome datasets in gnomAD, as well as the UK Biobank and DiscovEHR, which have >90% samples labeled as of European ancestry.

The online version of this article includes the following figure supplement(s) for figure 1:

**Figure supplement 1.** Exonic de novo mutation rates per generation per site estimated from a sample of 2976 parent-offspring trios data from *Halldorsson et al., 2019*, by mutation type.

**Figure supplement 2.** De novo mutation rate in exons in a sample of 2976 parent-offspring trios, by average methylation levels.

**Figure supplement 3.** Effect of trinucleotide context on mutation rate and mutation saturation at methylated CpG sites.

**Figure supplement 4.** Comparing the distribution of CpG transition rates at methylated sites within and ouside exons.

inside and outside exons, which differ in a number of these features, there is no appreciable difference in average DNM rates (Fisher Exact Test (FET) p-value = 0.1, *Figure 1—figure supplement 4a*). Similarly, the rate at which two DNMs occur at the same site, a summary statistic that reflects the variance in mutation rates, is not significantly different for methylated CpGs inside versus outside exons (FET p-value = 0.35; *Figure 1—figure supplement 4b*). Thus, while there is some variation in mutability per site among methylated CpGs, it appears to be small relative to the mean mutation rate across all methylated CpGs.

Considering all such CpG sites therefore, we ask what fraction are segregating at existing sample sizes. To this end, we collate polymorphism data made public by gnomAD (*Karczewski et al., 2020*), the UK Biobank (*Szustakowski, 2020*), and the DiscovEHR collaboration between the Regeneron Genetics Center and Geisinger Health System (*Dewey et al., 2016*) in order to ascertain whether both C and T alleles are present in a sample of ~390K individuals (Materials and methods).

To focus on the subset of genic changes most likely to be neutrally-evolving, we consider the ~350,000 methylated CpG sites at which C > T mutations do not change the amino acid. At these sites, 94.7 % of all possible synonymous CpG transitions are observed in the gnomAD data alone, and 98.8 % in the combined sample including all three datasets (*Figure 1*). In other words, nearly every methylated CpG site where a mutation to T is putatively neutral has experienced at least one such mutation in the history of the sample of 390K individuals. Even in the least mutable CpG trinucleotide

context, 98 % of putatively neutral sites are segregating in current samples (*Figure 1—figure supplement 3b*). These observations imply that in the absence of selection, almost every methylated CpG site would be segregating a T at current sample sizes--and further that not seeing a T provides strong evidence it was removed by selection.

## Testing a neutral model for individual sites

The mutation saturation at methylated CpG sites provides a robust approach to identify individual sites that are not neutrally-evolving. One way to view it is in terms of a p-value: under a null model with no selection, from which we assume that synonymous sites are drawn, all but 1.2 % of neutral sites are segregating in a sample of 390K individuals. Therefore, if a given non-synonymous site, say, is invariant in a sample of ≥390K individuals, we can reject the neutral null model for this site at a significance level of 0.012. Similarly, we can ask about the probability that an invariant non-synonymous site is neutral, using a false discovery rate (FDR) approach: given that 1.2 % of neutral sites are invariant, whereas 7.4 % of non-synonymous sites are, the FDR is 1.2/7.4 = 16%. Thus, at current sample sizes, there is a substantial amount of information about whether individual CpG transitions are deleterious. By contrast, in a smaller sample with only 10 % of putatively neutral sites segregating, there is almost no information about selection in observing individual sites to be invariant ($p \leq 0.9$).

This approach implicitly assumes that synonymous and non-synonymous sites do not differ in their distributions of mutation rates and that their distributions of genealogical histories are also the same, i.e.,, that the two types of sites are subject to comparable effects of linked selection. While we cannot examine whether the distributions of mutation rates are identical for lack of data, we verify that the mean de novo mutation rates do not differ for synonymous sites and for various non-synonymous annotations (*Figure 2a*); we also check that the distributions of methylation levels (conditional on ≥65%), an important determinant of mutation rates, are similar for synonymous and non-synonymous sites (with a significant but small shift towards higher methylation and thus presumably higher mutation rates for non-synonymous sites; *Figure 2—figure supplement 1*). In turn, the standard assumption of similar distributions of genealogical histories seems sensible, given that the sites are interdigitated within genic regions (*McDonald and Kreitman, 1991*). Under these few and at least somewhat testable assumptions, the approach based on mutation saturation at methylated CpG sites then enables us to directly pinpoint individual sites that are not neutrally evolving. We note further that if synonymous sites are not all neutral and instead some fraction are under selection, the same idea would apply, but the null model would have to be modified accordingly.

## Comparing the fraction of segregating sites across annotations

Under these same weak assumptions, it is also possible to compare the proportion of methylated CpG sites polymorphic for a transition across annotations. Here, we consider the fraction of sites segregating a transition in each annotation class in a sample of 780K chromosomes, rescaled by the fraction segregating at synonymous sites. All categories of missense, loss-of-function, and regulatory variants show a significant depletion in the fraction of segregating sites compared to synonymous variants (*Figure 2b*). The deficit for a given annotation is an indicator of the deleteriousness of de novo mutations in that annotation. Specifically, in our sample of 780K, the deficit for each annotation reflects sites for which we can reject neutrality at a significance level of 0.012.

These data therefore suggest that there are ~27 % fewer loss-of-function variants than would be expected under neutrality; at invariant sites within this annotation, neutrality can be rejected at an FDR of only 4.4 % ( = 1.2/27). A 27 % deficit of loss-of-function variants is again seen if we match the sites to synonymous mutational opportunities with the same predicted level of linked selection, i.e., with similar genealogical histories (*McVicker et al., 2009*; *Figure 2—figure supplement 2a*). Supporting the widely used assumption that LOF mutations within a gene are equivalent (after filtering for those at the end of transcripts; *Karczewski et al., 2020*; *Cassa et al., 2017*), when we compare the set of CpG sites at which mutations are annotated as leading to protein-truncation in the first versus the second half of transcripts, approximately the same number are missing mutations relative to synonymous sites in both subsets (*Figure 2—figure supplement 3*; FET p-value = 0.9). By comparison, the fraction of missense mutations and splice region variants not observed in current samples is only about 5.3%, and the FDR 22.6 % ( = 1.2/5.3) (whether or not we match for the effects of linked selection; see *Figure 2—figure supplement 2a*).

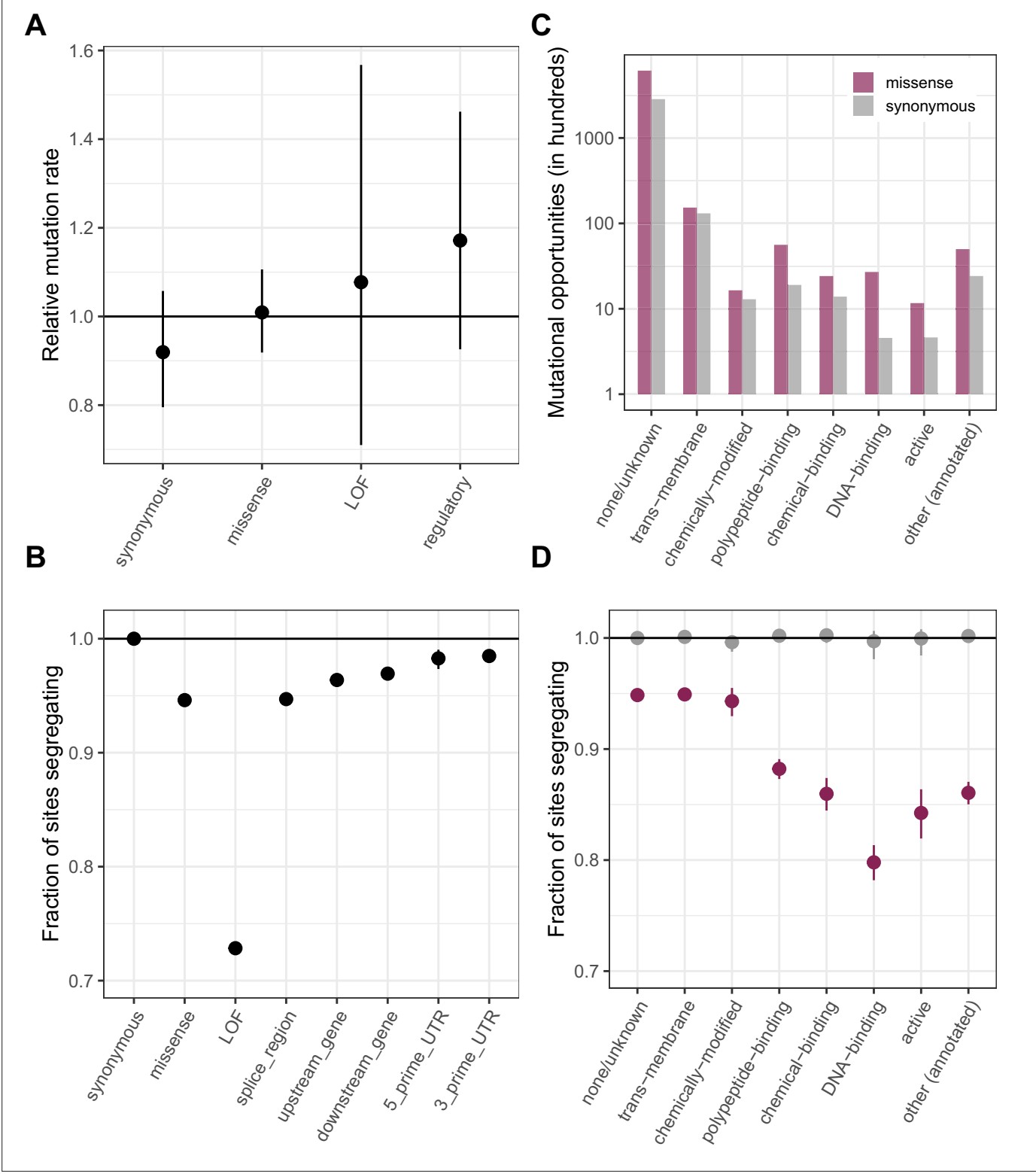

**Figure 2.** Comparing de novo mutation rates and the fraction of segregating sites across annotations. (**a**) DNM rates for CpG transitions at highly methylated sites by annotation class, rescaled by the total DNM rate in exons. Fisher exact tests (FETs) of the proportion of sites with DNMs in each annotation compared to all other annotations yield p-values > 0.1 in all cases. (**b**) Fraction of highly methylated CpG sites that are segregating as a C/T polymorphism in an annotation class, relative to the fraction of synonymous sites segregating. Error bars are 95 % confidence intervals assuming the number of segregating sites is binomially distributed (FET p-values << $10^{-5}$ for comparisons of all annotations with synonymous sites). LOF variants are

*Figure 2 continued on next page*

Figure 2 continued

defined as stop-gained and splice donor/acceptor variants that do not fall near the end of the transcript, and meet the other criteria to be classified as 'high-confidence' loss-of-function in the gnomAD data (see Materials and Methods). (**c**) The amount of data for synonymous and missense changes involving highly methylated CpG transitions by the type of functional protein site. (**d**) The proportion of synonymous and missense segregating C/T polymorphisms in different classes of functional sites. Error bars are 95 % confidence intervals assuming the number of segregating sites is binomially distributed (FET p-values << $10^{-5}$ for comparisons of all missense annotations with synonymous sites; Materials and methods). All annotations are obtained using the canonical transcripts of protein coding genes (see Materials and methods).

The online version of this article includes the following figure supplement(s) for figure 2:

**Figure supplement 1.** Distribution of methylation levels at synonymous and non-synonymous methylated CpG sites in testes and ovaries.

**Figure supplement 2.** The effect of background selection on the fraction of sites segregating in each annotation.

**Figure supplement 3.** Comparing LOF CpG transitions at methylated sites in exons that constitute the first vs. second halves of canonical protein coding transcripts.

**Figure supplement 4.** Comparing de novo mutation rates and the fraction of segregating sites across annotations obtained using the worst consequence in protein coding transcripts by predicted severity, instead of canonical transcripts as in *Figure 2*.

**Figure supplement 5.** De novo C>T mutation rates at methylated CpG sites and the fraction of sites segregating in CADD score bins.

While LOF and missense annotation classes are most commonly used in determinations of variant pathogenicity, any two sets of methylated CpGs with similarly-distributed mutation rates can be ranked in this manner. As one example, we stratify missense mutations by the type of functional site in which they occur. For the subset of sites at which missense mutations may disrupt or alter binding, particularly DNA-binding, there is a ~ 12–20% deficit in segregating sites relative to what is seen at synonymous sites, in contrast, say, to the much smaller deficit at missense changes within transmembrane regions (*Figure 2c–d*, *Figure 2—figure supplement 4*; *Figure 2—figure supplement 2b*). In other words, observing a DNA-binding missense site that is invariant provides stronger statistical evidence that it is deleterious than observing an invariant missense site with no additional functional information (e.g. the FDR is ~1/20 vs. ~1/5).

We can also check that the fraction of sites segregating is inversely proportional to the predicted functional importance of the sites using CADD scores (*Rentzsch et al., 2019*), widely used measures of constraint that incorporate functional annotations and measures of conservation. Across deciles, mean de novo transition rates at methylated CpGs are similar (*Figure 2—figure supplement 5a*) and, as expected, the fraction of segregating sites decreases with increasing CADD scores (*Figure 2—figure supplement 5b*). We note, however, that mutation rates may not always be similar across comparison groups: considering all CpG sites in exons (i.e. not only highly methylated ones), for example, de novo mutation rates are much more variable across CADD deciles (*Figure 2—figure supplement 5c*). Consequently, the depletion of segregating sites no longer has a simple interpretation (*Figure 2—figure supplement 5d*), instead reflecting a combination of differences in mutation rates and fitness effects. By implication, while CADD scores are meant to isolate the effects of selection, they will in some cases classify sites that have high mutation rates as less constrained, and vice versa.

## What can be learned about other mutation types?

Given that current exome samples are informative about selection on transitions at methylated CpGs, a natural question is to ask to what extent there is also information for less mutable types, with mutation rates on the order of $10^{-8}$ or $10^{-9}$ per site per generation. For sites with mutation rate on the order of $10^{-9}$, which is the case for the vast majority of non-CpGs, the fraction of possible synonymous sites that segregate in a sample of 780K chromosomes is very low: for instance, it is 5 % for T > A mutations, which occur at an average rate of $1.2 \times 10^{-9}$ (*Figure 1—figure supplement 1*) and 27 % even for other C > T mutations, which occur at a rate of $0.9 \times 10^{-8}$ per site (*Figure 1—figure supplement 1*), compared to ~99 % for C > T mutations at methylated CpGs (*Figure 3a*). For invariant sites of these less mutable types, there is little information with which to evaluate the fit to the neutral null in current samples. Reflecting this lack of information, in the p-value formulation, monomorphic sites would be assigned $p \leq 0.95$ for T > A sites and $p \leq 0.73$ for C > T sites.

How large samples have to be for other mutation types to reach saturation depends on the length of the genealogy that relates sampled individuals, i.e., the sum of the branch lengths, which corresponds

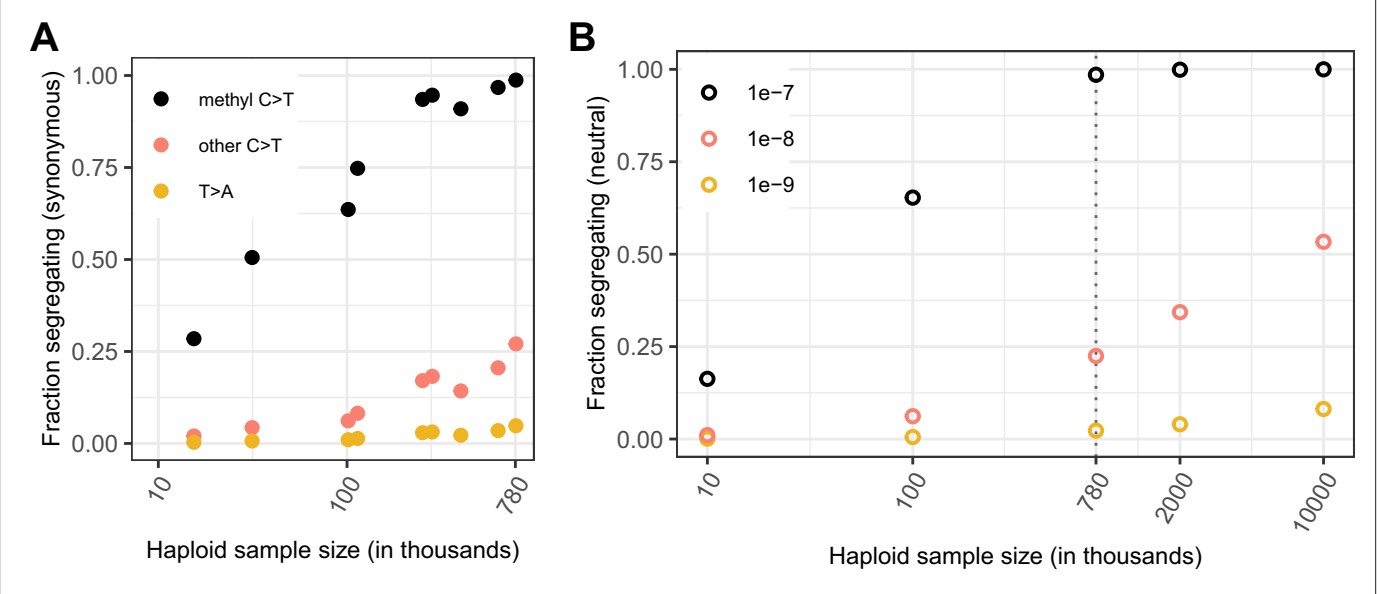

**Figure 3.** Comparing the fraction of sites observed and expected to be segregating under neutrality, by mutation type and sample size. (**a**) Fraction of possible synonymous C > T mutations at CpG sites methylated in the germline and at all other C sites, and the fraction of possible synonymous T > A mutations that are observed in a sample of given size. (**b**) Fraction of sites segregating in simulations, assuming neutrality, a specific demographic model and a given mutation rate (see Materials and methods).

The online version of this article includes the following figure supplement(s) for figure 3:

**Figure supplement 1.** The expected length of the genealogy under different demographic models and for varying sample sizes.

**Figure supplement 2.** Mutation saturation in bins of sites compared to single mCpG sites.

to the number of generations over which mutations could have arisen at the site. For a mutation that occurs at rate $1.17 \times 10^{-7}$ per generation, the average length of the genealogy would have to be greater than 8.5 million ($1/1.17 \times 10^{-7}$) generations for at least one such mutation to be expected at a site. That synonymous CpG sites are close to saturation when they experience mutations to T at this rate suggests that this is in fact the case. Indeed, given that more than one mutation has occurred at a substantial fraction of sites (**Karczewski et al., 2020**; **Harpak et al., 2016**), the average length of the genealogy relating the 390K individuals is expected to be substantially longer: about 39 million generations (calculated from the probability of at least one mutation under a Poisson distribution; see Materials and methods). The observation that mutation types with rates on the order of $10^{-8}$ are far from saturation further indicates that the average length of the genealogy for these 390K individuals is substantially shorter than 100 million generations. These rough calculations thus provide a sense of the length of the genealogical history represented by these 390K individuals.

To explicitly examine the relationship between sample size, mutation rate and the amount of variation at a locus, we simulate neutral evolution at a single site with the three different mutation rates above, under a variant of the widely-used Schiffels-Durbin demographic model for population growth in Europe (**Schiffels and Durbin, 2014**), in which we set the effective population size $N_e$ equal to 10 million for the past 50 generations (Methods). While this model is clearly an oversimplification, it recapitulates observed diversity levels for synonymous mutations reasonably well (**Figure 3**). Consistent with the rough estimate above, under our choice of demographic model, a sample of 780K chromosomes has a genealogy spanning an average of 34 million generations (**Figure 3—figure supplement 1a,b**).

From first principles, the length of the genealogy is expected to increase much more slowly than linearly with the number of samples (**Hudson, 1990**; **Nelson et al., 2012**). Indeed, increasing the number of samples by a factor of 12 only increases the average tree length ~3.3 x (**Figure 3b**, **Figure 3—figure supplement 1a,b**); thus, a site that mutates at rate $10^{-9}$ per generation is expected to have experienced ~0.04 mutations in the genealogical history of a sample of ~1 million, and 0.1

mutations in a sample of 10 million. The implication is that saturation for mutation rates of $10^{-8}$ or $10^{-9}$ per site per generation may not be achievable any time soon.

Quantitative predictions of our model are subject to the considerable uncertainty about the demographic history and in particular about the recent effective population size in humans (*Figure 3—figure supplement 1b*). Moreover, for simplicity, we model one or at most two populations, when samples that combine individuals from more diverse genetic ancestries have longer genealogical histories (*Figure 3—figure supplement 1c*; see *Figure 1*) and thus capture more variation. Perhaps most importantly, for the very large sample sizes considered here, the multiple merger coalescent is a more appropriate model (*Nelson et al., 2012*; *Bhaskar et al., 2014*). Nonetheless, the qualitative statement that less mutable types will remain very far from saturation in the foreseeable future should hold.

In the absence of information about single sites for most mutational types in the genome, it is still possible to learn to a limited degree about selection using bins of sites. If we construct a bin of $K$ synonymous sites with the same average mutation rate per bin as a single methylated CpG, then at least one site per bin is polymorphic in ~99 % of bins (see *Figure 3—figure supplement 2* for an example with T > A mutations and $K\sim100$), just as ~99 % of individual methylated CpG sites are segregating. Thus, if a bin of $K$ non-synonymous sites with the same average mutation rate is invariant, the p-value associated with the bin is 0.01, indicating that one or more sites in the bin is likely to be under selection.

## How strong is the selection that leads to invariant methylated CpG sites?

Leveraging saturation to identify a subset of sites that are not neutrally-evolving makes appealingly few assumptions, but provides no information about how strong selection is at those sites. To learn about the strength of selection consistent with methylated CpG sites being monomorphic, a series of strong assumptions are needed: we require a demographic model, a prior distribution on $hs$ and a mutation rate distribution across sites. Here, we assume a relatively uninformative log-uniform prior on the selection coefficient $s$ ranging from $10^{-7}$ to 1 and fix the dominance coefficient $h = 0.5$ (as for autosomal mutations with fitness effects in heterozygotes, we only need to specify the compound parameter $hs$; reviewed in *Fuller et al., 2019*), as well as a fixed mutation rate of $1.2 \times 10^{-7}$ per site per generation. We rely on the demographic model for population growth in Europe described above (*Schiffels and Durbin, 2014*); as is standard (*Sawyer and Hartl, 1992*; *Boyko et al., 2008*; *Williamson et al., 2005*; *Eyre-Walker et al., 2006*; *Kim et al., 2017*; *Cassa et al., 2017*; *Simons et al., 2014*), we also assume that $hs$ is fixed over time, even as the effective population size changes dramatically. Under these assumptions, we estimate the posterior distribution of $hs$ at a site, given that the site is monomorphic, segregating with 10 or fewer derived copies of the T allele, or segregating with more than 10 copies (*Figure 4a and b*, Methods). These posterior distributions are estimates of the DFE at an individual mCpG site conditional on seeing 0 copies, 1–10 copies or >10 copies of the T allele.

Because Bayes odds provide a natural way to summarize the strength of statistical evidence that comes from the observation at a single site, we consider the Bayes odds that a mutation is subject to $hs > 0.5 \times 10^{-3}$, i.e., is under strong selection (see Materials and methods). In small samples, in which most sites are monomorphic, being monomorphic is consistent with both neutrality and very strong selection (*Figure 4a*) and the Bayes odds are close to 1, reflecting the fact that the observation barely shifts our prior assumptions (*Figure 4b*). In contrast, with larger sample sizes, in which putatively neutral CpG sites reach saturation, the posterior distribution for invariant sites is highly peaked–what is not segregating is likely strongly deleterious–and accordingly the Bayes odds become substantially greater than 1. Notably, at current sample sizes of 390K individuals, there is still some dependence on the prior (*Figure 4—figure supplement 1*), but the Bayes odds of $hs > 0.5 \times 10^{-3}$ at an invariant methylated CpG are large. Given our choice of prior, the odds are 15:1 (*Figure 4b*), which suggests that most (~15/16) of the ~27 % of LOF mutations and ~6 % of missense mutations not seen in current samples are subject to this degree of selection.

While the relationship of selection strengths to clinical pathogenicity is not straight-forward, selection coefficients on that order are likely to be of relevance to determinations of pathogenicity in clinical settings (*Cassa et al., 2017*; *Kaplanis et al., 2020*). Indeed, mutations with $hs > 0.5 \times 10^{-3}$ may be highly deleterious to some individuals that carry them, enough to produce clinically visible effects,

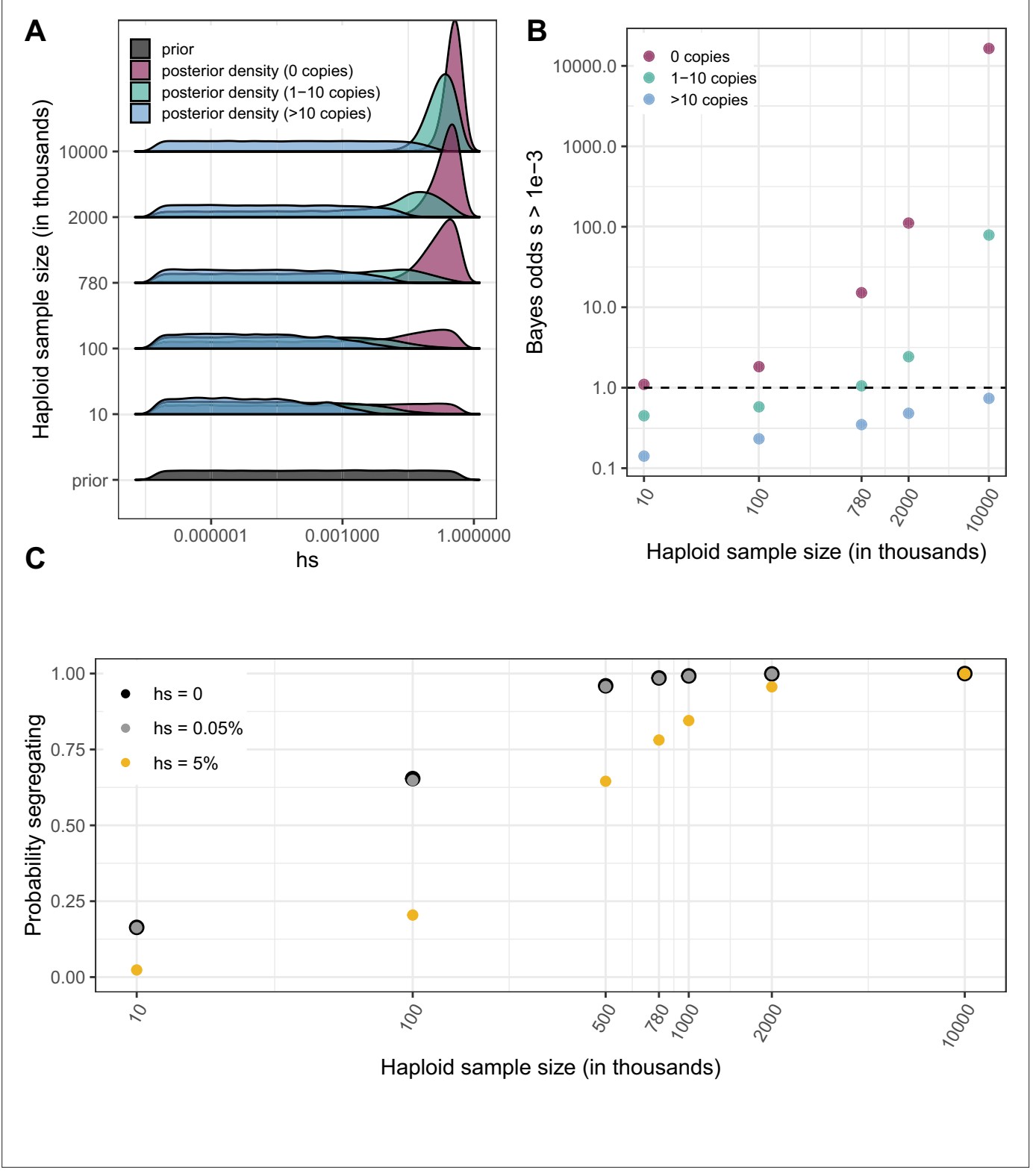

**Figure 4.** Quantifying the strength of selection associated with invariant and segregating sites. (**a**) Prior and Posterior log densities of *hs* for a C > T mutation at a methylated CpG site observed at 0, 1–10, or >10 copies at various sample sizes. (**b**) Bayes odds (i.e. posterior odds divided by prior odds) of *s* > 0.001 for a C > T mutation at a methylated CpG site observed at 0, 1–10, or >10 copies, at various sample sizes. (**c**) Probability of a methylated CpG site segregating a T allele in simulations, if the mutation has no fitness effects (hs = 0) and if it is deleterious (with a heterozygote selection coefficient hs = 0.05%) or highly deleterious (with a heterozygote selection coefficient hs = 5%).

*Figure 4 continued on next page*

*Figure 4 continued*

The online version of this article includes the following figure supplement(s) for figure 4:

**Figure supplement 1.** Effect of the choice of prior on Bayes odds of h$s$ > 0.5x10$^{-3}$.

**Figure supplement 2.** Odds of non-synonymous variants having been classified as pathogenic in ClinVar and DDD if they occur at sites that are either invariant (0 copies) or segregating ( > 0 copies) in a sample of 780 K chromosomes.

**Figure supplement 3.** For various sample sizes, prior and posterior log densities for *hs*, and the Bayes odds of *s* > 10$^{-3}$ (and *h* = 0.5) for a mutation observed at 0, 1–10, or >10 copies.

**Figure supplement 4.** Comparison of measures of deleteriousness at 1.1 million mutational opportunities for methylated CpG (mCpG) transitions vs. 90 million other mutational opportunities in exons.

**Figure supplement 5.** Estimating the DFE of LOF mutations using posterior densities of *hs* for invariant and segregating mCpG sites.

but vary substantially in their penetrance. Accordingly, mutations classified as pathogenic in ClinVar (*Landrum et al., 2018*) or identified as underlying severe developmental disabilities in the Deciphering Developmental Disorders (DDD) cohort (*Kaplanis et al., 2020*) are 6-fold to 51-fold enriched at sites invariant in 390K individuals compared to those classified as benign (*Figure 4—figure supplement 2*). This analysis comes with important caveats–notably that the classifications of pathogenicity rely in part on the presence or absence of mutations in reference databases–but it suggests an enrichment on par with estimated Bayes odds of strong selection.

## Discussion

### Interpreting polymorphic sites in current reference databases

In a sufficiently large sample, even a segregating site can be subject to strong selection (*Figure 4a and b*). For instance, in current exome sample sizes, a C > T mutation at a methylated CpG site with hs = $0.5 \times 10^{-3}$ is almost always observed segregating (*Figure 4c*). This follows from the expectation under mutation-selection-drift balance (*Gillespie, 1998*): in a constant population size, a mutation that arises at rate $1.2 \times 10^{-7}$ per generation and is removed by selection at rate *hs* = 0.05% per generation has an expected population frequency of $2.4 \times 10^{-4}$; in a sample of 780K, the mean number of copies is 187. Even with substantial variation due to genetic drift and sampling error, such a site should almost always be segregating at that sample size. In fact, even a mutation with *hs* of 5 % would quite often be observed. Thus, segregating sites in large samples are a mixture of neutral, weakly selected and strongly selected sites. An implication is that, although large reference repositories such as gnomAD were partly motivated by the possibility of excluding deleterious variants, as samples grow in size, it cannot simply be assumed that clinically relevant variants are absent from reference datasets. In principle, the only mutations never seen as samples grow in size would be the ones that are embryonically lethal.

More generally, any interpretation of variants of unknown function by reference to repositories such as gnomAD or disease cohorts enriched for deleterious variation (*Taliun et al., 2021*), whether the goal is to exclude benign variants or identify likely pathogenic ones, is implicitly reliant on assumptions that change with sample size and dramatically differ by mutation type. At current sample sizes, invariant methylated CpGs are likely highly deleterious; for less mutable types, the information content at invariant sites is very limited at even the largest sample sizes considered (*Figure 4—figure supplement 3*). Similarly, learning about the fitness consequences of segregating mutations from their observed frequencies is contingent on assumptions about the mutation rate, and the demographic history of the sample.

### The distribution of fitness effects in human genes

As we show, a typical site in the genome, with a mutation rate of 10$^{-8}$ per generation, does not provide much information about selection (*Figure 4—figure supplement 3*), because the average length of the genealogy is likely substantially less than 10$^{8}$ generations. One exception, which is a special case, is gene loss: each gene can be conceived of as a single locus at which many possible LOF mutations have the same fitness impact (*Karczewski et al., 2020*; *Cassa et al., 2017*; *Fuller et al., 2019*; *Weghorn et al., 2019*; Agarwal, Fuller, Przeworski, in prep.). The mutation rate to LOF, calculated by summing rates of individual LOF mutations, is ~10$^{-6}$ per gene per generation on average (*Karczewski*

*et al., 2020*), such that in the absence of selection, many LOF mutations are expected in most genes. At this special subset of sites, the distribution of fitness effects can be inferred by binning loss-of-function variants within genes (*Karczewski et al., 2020*; *Cassa et al., 2017*; *Weghorn et al., 2019*; Agarwal, Fuller, Przeworski, in prep.).

An analogous strategy to overcome sample size limitations at other types of sites is to infer selection in bins of sites (*Dukler et al., 2021*); however, if sites within a bin vary in their fitness effects, inferences based on these bins are not straight-forward. Indeed, the mutation frequency in a bin reflects the harmonic mean of *hs* across sites in the bin weighted by (unknown) mutation rates across sites (see Materials and methods).

Given these limitations, individual methylated CpG sites can provide a useful point of entry to understanding the DFE in humans. Although methylated CpG sites appear under somewhat less constraint than other sites, the differences are subtle (*Figure 4—figure supplement 4*), and what we learn at these sites can tell us what to expect more generally. As a first step, we can obtain a DFE across non-synonymous mCpG sites by weighting the densities for segregating and invariant sites (*Figure 4a*) by the proportion of sites in each category (an example for possible LOF mutations is shown in *Figure 4—figure supplement 5*, for sample sizes of 15K and 780K) . In current samples, the posterior odds for invariant methylated CpGs having hs $\geq 0.5 \times 10^{-3}$ are 92 % under our model, whereas they are 37 % for segregating methylated CpGs. Considering possible LOF mutations at methylated CpGs, of which 27 % are not observed in current samples, these odds imply that the fraction of de novo LOF mutations with hs $\geq 0.5 \times 10^{-3}$ is roughly 52 % ( $= 0.27 \times 0.92 + 0.73 \times 0.37$).

We can use a similar approach to estimate the minimum fraction of de novo mutations that lead to a deleterious non-synonymous change. For missense sites, given the same uninformative prior on *hs* as for LOF mutational opportunities, the fraction estimated to be highly deleterious is 40 % ( $= 0.05 \times 0.92 + 0.95 \times 0.37$). Since ~0.97 % of all de novo point mutations are missense and ~0.07 % lead to a LOF (see Methods), these estimates translate into roughly a 1 in 236 chance ( $= 40\% \times 0.97\% + 52\% \times 0.07\%$) that a de novo mutation has an effect *hs* $\geq 0.5 \times 10^{-3}$. Assuming, finally, that each individual inherits 70 new mutations (*Kong et al., 2012*; *Jónsson et al., 2017*), these estimates imply that one out of every 3.4 individuals is born with a new and potentially highly deleterious, non-synonymous mutation. This calculation is based on only two frequency categories, however, discarding the information contained in allele frequencies at segregating sites, and only point mutations are taken into account. Thus, the true fraction is likely substantially higher.

## Outlook

Moving forward, we should soon have substantial information not only about the DFE but the strength of selection at individual CpG sites (*Figure 4*). Inferences based on them, or indeed any sites, will need to rely on an accurate demographic model, particularly for the recent past of most relevance for deleterious mutations; this problem should be surmountable, given the tremendous recent progress in our reconstruction of population structure and changes in humans (*Schiffels and Durbin, 2014*; *Kelleher et al., 2019*; *Speidel et al., 2019*). Inferences will also require a good characterization of mutation rate variation across CpG sites, as is emerging from human pedigree studies and other sources (*Jónsson et al., 2017*; *Poulos et al., 2017*; *Vöhringer et al., 2020*; *Seplyarskiy and Sunyaev, 2021*), and careful consideration of the effects of multiple hits (*Harpak et al., 2016*) and biased gene conversion (*Glémin et al., 2015*). It will also be of interest to revisit the standard assumptions that go into inferring a DFE, including that all mutations are at least partially dominant in their fitness effects; that the DFE remains fixed even as the effective population size changes by orders of magnitude; and that the distribution is bounded above at 0, when some of the mutations segregating in large samples are likely to be weakly beneficial. Putting these elements together, robust inference of the fitness effects of mutations in human genes should finally be within reach, through the lens of CpG sites.

## Materials and methods
### Processing de novo mutation data

We focused on ~190,000 published de novo mutations in a sample of 2976 parent-offspring trios that were whole genome sequenced (*Halldorsson et al., 2019*). To date, this is the largest publicly available set of trios that, to our knowledge, have not been sampled on the basis of a disease phenotype.

Unless otherwise specified, we used these DNMs to calculate mutation rates, as described in later sections. We converted hg38 coordinates to hg19 coordinates using UCSC Liftover. We excluded indels, and all DNMs that occur outside the ~2.8 billion sites covered by gnomAD v2.1.1 whole genome sequences. We obtained the immediately adjacent 5' and 3' bases at each position from the hg19 reference genome, so that we had each de novo mutation within its trinucleotide context; we used this information to identify CpG sites. Where such data were available (for 89 % of CpG de novo mutations), we also annotated each site with its methylation status measured by bisulfite sequencing in testis sperm cells and ovaries (see *Appendix 1—table 1*).

We annotated DNMs with their variant consequences using Variant effect predictor (v87, Gencode V19) and the hg19 LOFTEE tool (*Karczewski et al., 2020*) to flag high-confidence ('HC') loss-of-function variants. We obtained the fraction of DNMs in the genome that occured at sites annotated as missense or LOF in the 'canonical' protein-coding transcript for each gene provided by Gencode.

## Processing polymorphism data

We downloaded publicly available polymorphism data from gnomAD (*Karczewski et al., 2020*), the UK Biobank (*Szustakowski, 2020*), the DiscovEHR collaboration between the Regeneron Genetics Center and Geisinger Health System (*Dewey et al., 2016*), and 1000 Genomes Phase 3 (*Auton et al., 2015*). Where needed, we lifted over coordinates to the hg19 reference assembly using the UCSC LiftOver tool. Salient characteristics of these samples are as follows:

| Dataset | Regions Sequenced | Individuals | Variants | Populations sampled | Original alignment |
|---|---|---|---|---|---|
| 1000 genomes Phase 3 (also included in gnomAD) | Genomes | 2504 | 84 million | mixture | hg19-b37 |
| gnomAD v2.1.1 | Exomes | 125,748 | 15 million | mixture | hg19 |
| gnomAD v2.1.1 | Genomes | 15,708 | 230 million | mixture | hg19 |
| UK Biobank | Exomes | 199,932 | 16 million | ~93 % European ancestry | hg38 |
| DiscovEHR | Exomes | 50,726 | 8 million | ~98 % European ancestry | hg19-b37 |

For the gnomAD data, we obtained the allele frequency for each variant in the full exome and genome samples, as well as their Non-Finnish European ('NFE') subsets from the VCF files (in hg19 coordinates) provided. For each sample, we obtained the set of segregating sites (i.e. the set of variants that pass gnomAD quality filters and have an allele frequency >0 in the sample). For the 1000 Genomes Phase-3 data, we obtained the set of variant positions similarly. Note that the 1000 Genomes samples are also contained within the gnomAD sample. For the DiscovEHR sample, allele frequencies are available where MAF >0.001 (and set equal to 0.001 for lower values > 0); we can thus determine the set of sites segregating in this sample, but we do not have access to any other information about individual variants.

For the UK Biobank exome sequencing data, additional processing was required. We downloaded the population-level plink files with exome-wide genotype information for ~200,000 individuals. We excluded exome samples that did not pass variant or sample quality control criteria in the previously released genotyping array data. Specifically, we excluded samples that have a discrepancy between reported sex and inferred sex from genotype data, a large number of close relatives in the database, or are outliers based on heterozygosity and missing rate, as detailed in *Bycroft et al., 2018*. Finally, we excluded individuals who withdrew from the UK Biobank by the end of 2020. This left us with 199,932 exome samples that overlap with the high-quality subset of the genotyped samples. We additionally limited our analysis to the list of ~39 million exonic sites with an average of 20X sequence coverage provided by UK Biobank (*Szustakowski, 2020*). We transformed the processed plink files into the standard variant call format, polarized variants to the hg38 reference assembly, and obtained the frequency of the non-reference allele in the sample. We then lifted over the coordinates from hg38 to hg19 using the UCSC LiftOver tool. We excluded the few positions where the reference alleles were mismatched or swapped between the two assemblies.

All but 12 % of segregating mCpG transitions were shared across at least two non-overlapping datasets. Of segregating variants seen in one of the gnomAD or UK Biobank datasets, all but two variants had at least 5 % of individuals (and typically on the order of ~100 K) sequenced at that position.

Thus, we think it highly unlikely that we misclassified invariant sites as segregating, or vice versa. For ~9000 variants that are seen only once in the GHS data, we unfortunately did not have access to variant quality metrics. Excluding these sites only very slightly affects our results and does not change any qualitative conclusions.

## Identifying and annotating mutational opportunities in the exome

For all possible mutational opportunities in sequenced exons, we collated a variety of functional annotations. To this end, we first generated a list of all possible SNV mutational opportunities in the exome. We obtained the list of sites that fall in exons or within 50 base pairs (bp) of exons in Gencode v19 genes and that are among the ~2.8 billion sites covered by gnomAD v2.1.1 whole genome sequences. For each position, we extracted the reference allele from the hg19 assembly and generated the three possible single-nucleotide derived alleles. We also obtained the immediately adjacent 5' and 3' bases at each position from the hg19 reference genome, so that we had each mutational opportunity within its trinucleotide context; we used this information to identify CpG sites. Where such data were available, we also annotated each site with its methylation status in testis sperm cells and ovaries.

To identify sites at which variants or de novo mutations could be confidently assayed by whole-exome sequencing methods, we obtained regions targeted in whole exome sequencing from gnomAD and the UK Biobank. We limited our analysis to sites that were covered at 20X or more in the exome sequencing subsets of both gnomAD and UK Biobank (that lifted over correctly to the hg19 assembly), which we refer to as 'accessible sites'.

We then annotated the ~90 million mutational opportunities (at 30 million sites) with CADD scores and variant consequences using Variant effect predictor (v87, Gencode V19) and the hg19 LOFTEE tool (*Karczewski et al., 2020*) to flag high-confidence ('HC') loss-of-function variants. For loss-of-function variants, we also noted their location in the gene by exon number (e.g. in exon 10 of 12 exons in the gene). We used a published database of curated protein features derived from Refseq proteins (*Stanek et al., 2020*) to annotate all sites in protein coding genes that were associated with a particular type of functional activity (detailed functional annotations were available for about 60,000 of 1.1 million methylated CpG sites). At each site, we used either the primary 'site-type' annotation, or when that was missing or listed as 'other', we extracted the annotation from the more detailed 'notes' field where this information was provided.

Because there are multiple transcripts for each variant, we limited our analysis to the 'canonical' protein-coding transcript for each gene provided by Gencode to obtain a single annotation for each variant. For 10–20% of variants, this approach still yielded multiple possible consequences per variant, for instance, where there are multiple canonical transcripts due to overlapping genes. For these cases, we assigned one of the 'canonical' transcripts to the variant at random, to avoid making assumptions about their relative importance. Further overlaps within the same gene, for example, a missense variant that is also a splice variant in the same transcript, or a DNA-binding site that also undergoes a particular post-translational modification, were resolved in the same manner.

As an alternative approach, we obtained the worst consequence in all protein-coding transcripts for each variant, using the ranks of variant consequences by severity provided by Ensembl (*Appendix 1—table 1*). In the absence of systematic ranking criteria for the protein function annotations, we used the following order: sites that were designated as having catalytic activity ('active' sites) were given highest priority in overlaps, followed by DNA-binding sites, followed by other types of binding (to metal, polypeptides, ions), and finally by sites that are known to undergo post-translational or other regulatory modifications, and trans-membrane sites. Thus, a transmembrane site with regulatory activity is classified as a regulatory site, while a regulatory site with DNA-binding activity is classified as DNA-binding. Using these alternate criteria to group sites does not affect our conclusions (*Figure 2—figure supplement 4*).

All sources of annotation data are listed in *Appendix 1—table 1*. A list of CpG sites and annotations is provided as additional data.

## Comparing fitness effects across sets of mutational opportunities

To assess whether the set of 1.1 million C > T mutational opportunities at methylated CpG sites are systematically different from the other ~90 million exonic mutational opportunities in their potential fitness effects, we compared the distribution of CADD scores in the two groups using a

Kolmogorov-Smirnov test. We note that this comparison is likely to be somewhat confounded by differences in mutation rates, given our finding that CADD scores do not perfectly isolate the effects of selection from those of variability in mutation rates (*Figure 2—figure supplement 5c*). Since the mutation rate for methylated CpG sites is higher than for other types, this may lead them to appear somewhat less constrained than they actually are.

We further compared the fraction of C > T mutational opportunities at methylated CpGs in an annotation class vs. the fraction of other mutational opportunities in that class. We used a Fisher exact test (with a Bonferroni correction for four tests) to determine whether the two sets of mutational opportunities were differently distributed across synonymous, missense, regulatory, and LOF variant classes.

## Obtaining mean de novo mutation rates by mutation type and annotation

We counted the total number of de novo mutations in sequenced exons (~91 million mutational opportunities) for eight classes of mutations: two transitions and a transversion each at C and T sites, transitions at CpG sites with relatively low levels of methylation (defined here as <65%), and transitions at CpG sites with high levels of methylation ( ≥ 65%). To obtain the mutation rate per site per generation, we divided the counts by the haploid sample size (2 × 2976 individuals; see section 1) and the number of mutational opportunities of each type. We report 95 % confidence intervals assuming a Poisson distribution for mutation counts. The rates obtained (*Figure 1—figure supplement 1*) are similar to previous ones (*Kong et al., 2012*; *Jónsson et al., 2017*; *Gao et al., 2019*) and roughly consistent with the rates predicted by the gnomAD mutation model (*Karczewski et al., 2020*).

To evaluate the impact of methylation status on the mutation rate at CpG sites, we obtained the mean mutation rate for C > T mutations at CpG sites in each methylation bin as described above, separately for methylation levels in ovaries and testes. While there is a limited amount of data, especially for some low-methylation bins, our choice of cutoff for 'methylated' seems sensible (*Figure 1—figure supplement 2*).

We then calculated the mean mutation rate for methylated CpG transitions, for different compartments in the genome, namely in (a) exons and non-exons, (b) four variant consequence categories: synonymous, missense, regulatory, and LOF variants, (c) CADD score deciles, and (d) in exons that constitute the first half vs the second half of genes. We also calculated the mean mutation rate for methylated CpG transitions in four trinucleotide contexts (ACG, CCG, GCG, and TCG). In each case, we obtained the total number of de novo mutations and the Poisson 95 % confidence interval around mutation counts in each group, and divided by the number of mutational opportunities in the group. We tested if the proportion of methylated CpG sites with de novo C > T mutations in each non-synonymous compartment was different from the proportion of synonymous methylated CpGs with de novo C > T mutations, accounting for multiple tests.

## Variance in mutation rate at methylated CpGs

Although current samples of DNM data are large enough to compare the mean mutation rate at methylated CpGs across the annotation classes examined here, there is not enough data to directly compare variances in mutation rates. To learn how much broad scale features beyond methylation and the immediate trinucleotide context shape variation in mutation rates at methylated CpGs, we therefore relied on a broader set of regions for example those that fall inside and outside exons. Exonic and non-exonic regions differ considerably in epigenetic features and replication timing (*Stamatoyannopoulos et al., 2009*); yet, there is no discernable difference in average de novo mutation rates at methylated CpGs inside and outside sequenced exons (FET p-value = 0.10, *Figure 1—figure supplement 4a*). We also compared the number of double and single de novo hits in exons and non-exons using a Fisher exact test (p-value = 0.35, *Figure 1—figure supplement 4b*). Since the number of double hits reflects the variance in mutation rates across sites, these results lend some support to there being limited variation due to broad scale genomic features in transition rates at methylated CpGs.

## Calculating the fraction of sites segregating by annotation

For each methylated CpG site in the exome, there are three mutational opportunities (C > A, C > G, C > T); we focused only on the opportunities for C > T mutations. For each methylated CpG site then, we noted whether or not it was segregating, or in other words if there was a C > T variant in samples of individuals from gnomAD (*Karczewski et al., 2020*), the UK Biobank (*Szustakowski, 2020*), the DiscovEHR dataset (*Dewey et al., 2016*), and 1000 Genomes Phase 3 (*Auton et al., 2015*), processed as described above, or a combined sample of 390 K non-overlapping individuals.

Within the set of methylated CpG sites where C > T mutations are synonymous, we calculated the fraction segregating in each sample of interest. Similarly, for different subsets of methylated CpGs, namely those in (a) four variant consequence categories: synonymous, missense, regulatory, and LOF variants, (c) CADD score deciles, (d) functional site categories (e.g. trans-membrane vs catalytic sites in proteins), and (e) the first half vs the second half of genes, we calculated the fraction segregating in the combined sample of 390 K individuals. We rescaled the fraction of sites segregating in each annotation by the fraction of synonymous sites segregating in the sample.

We verified that the differences in the fraction of sites segregating across annotations are not due to variable impacts of linked selection across annotations. To do so, we calculated the fraction of sites segregating with sites in different annotations matched for B-statistics *McVicker et al., 2009*; we obtained very similar results with this approach (*Figure 2—figure supplement 2*).

We assumed that conditional on the number of mutational opportunities and a fixed probability of segregating for each site in a compartment, the number of sites segregating is binomially distributed, and obtained 95 % confidence intervals on that basis. We tested if the proportion of sites segregating in each compartment is different from the proportion segregating at putatively neutral (here, synonymous) sites using a Fisher exact test, accounting for multiple tests.

We also calculated the fraction of other types of synonymous sites segregating in each sample size of interest (specifically, for T > A variants, and C > Ts not at methylated CpG sites).

## Frequency of mutant alleles in bins of *K* sites

Within each annotation of interest, with an average mutation rate of *u* per site, we construct bins of *k* sites, such that *k* = *U/u*, where *U* is the mean mutation rate of a transition at methylated CpG site in that annotation class. The mean mutation rates are calculated for each mutation type within each annotation, as described in Section five above. We then count the fraction of bins in which no such mutations are observed. As an example, for T > A mutations, *k* is on the order of 100 (*Figure 3—figure supplement 2a*).

Since each bin can be treated as being comparable to a single neutral methylated CpG site, bins that contain only neutral sites are expected to contain at least one mutation in 99 % of bins; this is indeed the case for bins of synonymous sites (*Figure 3—figure supplement 2b*).

When considering sites that contain a mixture of neutral and selected sites, bins of *k* sites are no longer as readily comparable to methylated CpG sites, however (*Figure 3—figure supplement 2c*). If sites within a bin are under varying degrees of selection, then the mutation count reflects the harmonic mean of the strength of selection acting on individual sites. Specifically, under a deterministic model of mutation-selection balance, if $q_i$ is the allele frequency at the $i^{th}$ site in a bin of *k* sites:

$$q_{bin} = \sum_{i=1}^{k} q_i$$

then

$$q_{bin} = \sum_{i=1}^{k} \frac{u_i}{hs_i}$$

Assuming $u_i = u = U/k$,

$$q_{bin} = \sum_{i=1}^{k} \frac{(U/k)}{hs_i} = \left(\frac{U}{k}\right) \cdot \sum_{i=1}^{k} \frac{1}{hs_i}$$

that is, $q_{bin}$ is a function of the harmonic mean of *hs* at the *k* sites.

## Forward simulations

We used a forward simulation framework initially described in *Simons et al., 2014*, and modified in *Fuller et al., 2019*. Briefly, we modeled evolution at a single non-recombining bi-allelic site, which undergoes mutations each generation at rate $2N_eu$ in a panmictic diploid population of effective population size $N_e$. Each generation is formed by Wright-Fisher sampling with selection, where fitness is reduced by $hs$ in heterozygotes and $s$ in homozygotes for the T allele. We fixed the dominance coefficient $h$ as 0.5, and we chose one value of the selection coefficient $s$ from a log-uniform prior ranging from $10^{-7}$ to 1 for each simulation (for autosomal mutations with fitness effects in heterozygotes, we only need to specify the compound parameter $hs$; reviewed in *Fuller et al., 2019*). Given a mutation rate and a demographic model that specifies $N_e$ in each generation, we simulated the evolution of a site forward in time to determine whether the site is segregating in a sample of size $n$ at present.

We used $u = 1.2 \times 10^{-7}$ per site per generation to model CpG> TpG mutation at a methylated CpG site. The simulation framework allows for recurrent mutations, which are expected to arise often at this mutation rate. We also allowed for TpG> CpG back mutations at the rate of $5 \times 10^{-9}$ (calculated from de novo mutation data, as CpG> TpG mutations). To model T > A mutations, we used $u = 1.2 \times 10^{-9}$ per site per generation, with a back mutation rate of $1.2 \times 10^{-9}$ per site per generation; for C > T mutations not at methylated CpG sites, we used $u = 0.9 \times 10^{-8}$ per site per generation, with a back mutation rate of $5 \times 10^{-9}$ per site per generation (*Figure 1—figure supplement 1*). We note that, since the mutation rate increases with paternal and maternal ages, an implicit assumption is that the distribution of parental ages in the trio data is representative of the parental ages over the evolutionary history of exome samples.

For the demographic model, we relied on the Schiffels-Durbin model for population size changes in Europe over the past ~55,000 generations, preceded by a ~ 10 $N_e$ generation burn-in period of neutral evolution at an initial population size $N_e$ of 14,448 following ref (*Simons et al., 2014*). In the last generation, that is at present, we sampled $n$ individuals from the simulated population, to match the size of the sample of interest.

We calculated the probability that a site with the fixed mutation rate $u$ is segregating for a given value of $hs$ (with $s = 0$ under neutrality) as the proportion of simulations with those parameters in which the site is segregating for different sample sizes at present.

In comparing the output of these simulations to data, we considered two scenarios where we may either undercount or overcount segregating CpG sites in the data relative to the simulations. First, because we conditioned on the human reference allele being a CpG in data, we did not count sites where the CpG is the ancestral but not the reference allele. To check how often this is expected to occur, we mimicked this scenario in simulations, sampling a single chromosome at the end of the simulation as the mock haploid reference genome. The proportion of simulations in which CpG is the ancestral but not the reference allele is ~0.1%, that is, approximately the heterozygosity levels in humans. The second case is that for a subset of the CpG> TpG variants observed at present, the CpG mutation is the reference allele but is not ancestral. To mimic this scenario in our simulations, we simulated a site that starts as TpG (with a mutation rate of $5 \times 10^{-9}$ to CpG, and a back mutation rate ~$1.2 \times 10^{-7}$ to TpG) forward in time. Then, as above, we drew a single chromosome from the sample at the end of the simulation and set it as the reference. We obtained the proportion of simulations in which the C allele is the reference, starting from a TpG background. Reassuringly, this occurs in only 0.0014 % of simulations. We note that there is in principle a third scenario to consider, in which ApG or GpG sites is ancestral and a C/T polymorphism is found in the sample at present as a result of two mutations, one to T and one to C. Given the various mutation rates involved (all less than $5 \times 10^{-9}$), this double mutation case will be even less likely than the one in which TpG was ancestral. These rare scenarios should not have any substantive effect on our comparison of data to simulations, particularly when we only used such comparisons to examine qualitative trends.

## Inferring the strength of selection in simulations

We proposed $s$ from a prior distribution (with $h$ fixed at 0.5) and inferred the posterior distribution of $hs$ for a site with a T allele at 0 copies using a simple Approximate Bayesian Computation (ABC) approach. Specifically, we proposed $s$ such that $\log_{10}(s) \sim \text{Uniform}(-7,0)$; we simulated expected T allele counts under our model for 10 million proposals from the prior. We accepted the subset of the proposed values of $s$ where simulations yield 0 copies of the T allele in the sample at present; this

set of $s$ values is a sample from the posterior distribution of $s$ given that the site is monomorphic. We calculated the Bayes odds of $s > 10^{-3}$ as the ratio of the posterior odds of $s > 10^{-3}$ and the prior odds of $s > 10^{-3}$:

$$\frac{p\left(hs > 0.5 \times 10^{-3} \mid copies\, of\, T = 0\right) / p\left(hs \leq 0.5 \times 10^{-3} \mid copies\, of\, T = 0\right)}{p\left(hs > 0.5 \times 10^{-3}\right) / p\left(hs \leq 0.5 \times 10^{-3}\right)}$$

We similarly obtained posterior distributions of $hs$ for sites that are segregating at 0, 1–10 copies, or >10 copies, in samples of different sizes, and for three different choices of priors on $s$, namely: $s \sim$ Beta($\alpha = 0.001$, $\beta = 0.1$); log($s$)$\sim$ N(–6,2); and $N_e s \sim$ Gamma(k = 0.23, $\theta$ = 425/0.23), with $N_e$ = 10,000, based on the parameters inferred in **Eyre-Walker et al., 2006**. These are shown in **Figure 4—figure supplement 1**.

## Calculating odds of being pathogenic in ClinVar and DDD

We downloaded de novo mutation data for ~35 K individuals with developmental disorders (**Kaplanis et al., 2020**). We also obtained a list of 380 'consensus' genes from the same study; for these genes, there is evidence from multiple sources that LOF or missense mutations are causal in developmental disorders, such that they are used as part of diagnostic criteria in the clinic.

We downloaded ClinVar variants and excluded those that were not associated with at least one disease. We obtained the 'CLNSIG' annotation, which classifies each variant as benign or likely benign, pathogenic or likely pathogenic, or as having uncertain status or conflicting evidence.

We limited both DDD and ClinVar variants to non-synonymous C > T mutations at the subset of methylated CpG sites considered. Using variants in ClinVar and DDD at sites that are invariant in our sample of 780K, we calculated the odds that an invariant site is pathogenic (vs. benign) as follows:

$$\frac{p\left(pathogenic \mid copies\, of\, T = 0\right) / p\left(benign \mid copies\, of\, T = 0\right)}{p\left(pathogenic\right) / p\left(benign\right)}$$

where p(pathogenic) refers to the proportion of sites classified as such, and p(benign) is defined analogously.

In DDD, we considered mutations that fall in 380 consensus genes 'pathogenic', and mutations in all other genes benign; thus our 'benign' category likely contains some genes in which mutations are in fact pathogenic. In ClinVar, variants classified as 'pathogenic' or 'likely pathogenic' are assumed to be pathogenic; these are compared to two sets of benign variants, one limited strictly to variants classified in ClinVar as 'benign' or 'likely benign', and the other inclusive of variants for which the evidence is uncertain or inconclusive. The results are shown in **Figure 4—figure supplement 2**.

We note that since both ClinVar classifications and the identification of consensus genes in DDD rely in part on whether a site is segregating in datasets like ExAC, the degree of enrichment in **Figure 4—figure supplement 2** is hard to interpret.

## Calculating the average length of the genealogy of a sample in which methylated CpGs are saturated

Methylated CpG sites experience mutations to T at the rate of $1.17 \times 10^{-7}$ per generation; 99 % of such sites are segregating in a sample of 390 K individuals. Then the average length ($L$) of the genealogy relating the 390 K individuals can be calculated from the probability under a Poisson distribution of at least one mutation at 99 % of sites as $1-\exp(-1.17 \times 10^{-7} \times L) = 0.99$, which gives $L$ = 39 million generations.

## Coalescent simulations to obtain the length of genealogy of large samples

We simulated the genealogy of a sample of varying sizes using *msprime* (**Kelleher et al., 2016**) under different demographic histories, modifying the standard Schiffels-Durbin model (**Schiffels and Durbin, 2014**) as follows:

a. Demographic history for a sample of Utah residents with Northern and Western European ancestry (CEU) over 55,000 generations, with a recent $N_e$ of 10 million for the past 50 generations, described above.
b. CEU demographic history for 55,000 generations with a recent $N_e$ of 100 million for the past 50 generations.
c. CEU demographic history for 55,000 generations with 4.5 % exponential growth for the past 196 generations, with a current $N_e$ of ~100 million.
d. Demographic history for a sample of Yoruba sampled in Nigeria (YRI) from *Schiffels and Durbin, 2014*, modified with a recent $N_e$ of 10 million for the last 50 generations.
e. A structured sample from two populations that derived from an ancestral population with YRI demographic history 2000 generations ago, with YRI and CEU demographic histories respectively since, and a recent $N_e$ of 10 million for the last 50 generations in each.

In each case, we recorded the mean genealogy length over 20 iterations.

The code for implementing these different demographic models in *msprime* is available on the project github repository.

## Acknowledgements

We thank Peter Andolfatto, Kelley Harris, Hakhamanesh Mostafavi, Magnus Nordborg, Itsik Pe'er, Jonathan Pritchard, Guy Sella, as well as Arbel Harpak, Zach Fuller, and other members of the Andolfatto, Przeworski and Sella labs for helpful discussions. This work was supported by NIH grants GM121372 and GM122975 to MP.

## Additional information

### Competing interests

Molly Przeworski: Senior editor, *eLife*. The other author declares that no competing interests exist.

### Funding

| Funder | Grant reference number | Author |
| --- | --- | --- |
| National Institutes of Health | GM122975 | Molly Przeworski |
| National Institutes of Health | GM121372 | Molly Przeworski |

The funders had no role in study design, data collection and interpretation, or the decision to submit the work for publication.

### Author contributions

Ipsita Agarwal, Conceptualization, Data curation, Formal analysis, Investigation, Methodology, Visualization, Writing - original draft, Writing - review and editing; Molly Przeworski, Conceptualization, Investigation, Methodology, Project administration, Resources, Supervision, Writing - original draft, Writing - review and editing

### Author ORCIDs

Ipsita Agarwal http://orcid.org/0000-0001-8537-0008
Molly Przeworski http://orcid.org/0000-0002-5369-9009

### Decision letter and Author response

Decision letter https://doi.org/10.7554/eLife.71513.sa1
Author response https://doi.org/10.7554/eLife.71513.sa2

## Additional files

### Supplementary files
• Transparent reporting form

### Data availability
All source data are freely available to researchers, with sources provided in the manuscript and summarized in Appendix 1 - Table 1. Source files and code to generate the figures, and additional files containing the annotated set of CpG sites analysed in this manuscript, are available at https://github.com/agarwal-i/cpg_saturation (copy archived at https://archive.softwareheritage.org/swh:1:rev:a36cb87c0af373e81eae1935ba710c4417c46f69).

The following previously published datasets were used:

| Author(s) | Year | Dataset title | Dataset URL | Database and Identifier |
|---|---|---|---|---|
| Karczewski KJ | 2020 | gnomAD | https://gnomad.broadinstitute.org/downloads | gnomAD v2.1, v2.1 |

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

# Appendix 1

**Appendix 1—table 1.** List of data sources.

| Annotation type | Source |
| --- | --- |
| Exon coordinates | http://ftp.ebi.ac.uk/pub/databases/gencode/Gencode_human/release_19/gencode.v19.annotation.gtf.gz |
| Exon annotations | Variant Effect Predictor (VEP) v87 using Gencode v19 Ranks:https://m.ensembl.org/info/genome/variation/prediction/predicted_data.html |
| WGS covered regions and exome target regions (gnomAD v2.1.1) | https://gnomad.broadinstitute.org/downloads |
| Exome target regions (UK Biobank) | https://biobank.ndph.ox.ac.uk/ukb/ukb/auxdata/xgen_plus_spikein.GRCh38.bed (liftovered to hg19) |
| CpG methylation Testis | GEO Accession GSM1127119 (https://www.ncbi.nlm.nih.gov/geo/) |
| CpG methylation Ovary | GEO Accession GSM1010980 (https://www.ncbi.nlm.nih.gov/geo/) |
| CADD | CADD v1.4 (https://cadd.gs.washington.edu/download) |
| B-statistic | https://doi.org/10.1371/journal.pgen.1000471 (lifted over to hg19) |
| Functional site annotations | https://ftp.ncbi.nlm.nih.gov/refseq/H_sapiens/mRNA_Prot/ and https://www.prot2hg.com |
| De novo mutations | Decode: https://doi.org/10.1126/science.aau1043 (Data S5) DDD: https://doi.org/10.1038/s41586-020-2832-5 (Supp. Table 1) |
| Polymorphism data | gnomAD: https://gnomad.broadinstitute.org/downloads UK Biobank: https://biobank.ctsu.ox.ac.uk/showcase/field.cgi?id=23155 DiscovEHR: http://www.discovehrshare.com/downloads 1,000 Genomes: ftp://ftp.1000genomes.ebi.ac.uk/vol1/ftp/release/20130502/ |
| ClinVar | https://ftp.ncbi.nlm.nih.gov/pub/clinvar/vcf_GRCh37/clinvar.vcf.gz |

