## [Decision Letter]

**Decision letter after peer review:**

Thank you for submitting your article "Mutation saturation for fitness effects at human CpG sites" for consideration by *eLife*. Your article has been reviewed by 3 peer reviewers, one of whom is a member of our Board of Reviewing Editors, and the evaluation has been overseen by Patricia Wittkopp as the Senior Editor. The reviewers have opted to remain anonymous.

The reviewers have discussed their reviews with one another, and the Reviewing Editor has drafted this to help you prepare a revised submission. All three reviewers were very enthusiastic about the manuscript, and with some revision it will clearly be suitable for publication in *eLife*.

The main revision reviewers agreed would be important to incorporate is the issue of multiple testing (highlighted by Reviewer #2).

While all the reviewers were enthusiastic about the manuscript as written, and none felt that additional analyses were necessary, there were a number of additional analyses suggested that reviewers felt could strengthen the paper if they were relatively straightforward to incorporate. These included:

1) Comparison of the nonsynonymous genome-wide DFE from smaller samples to the 780K, perhaps only focusing on methylated nonsynomous sites.

2) Comparison of overlap between these invariant sites and Clinvar or databases of patients with known developmental disorders.

3) Comparison of results under a different parameterization of human demography.

Reviewers had a number of additional suggestions that we felt would improve the manuscript. Perhaps first among these was a general feeling of "to be continued" in the discussion with multiple citations to another manuscript in prep. While reviewers were not against this strategy by any means, efforts to address the novelty of the current paper in the discussion (rather than simply hinting at exciting results in the forthcoming work) would be worthwhile.

Please do review and respond to the individual reviewer comments as well.

*Reviewer #1:*

Agarwal and Przeworski have performed a very timely and interesting study of the distribution of fitness effects (DFE) of new mutations. This study is timely because modern human population genetic datasets have finally achieved sample sizes for which a certain class of nucleotide sites, i.e. methylated CpG sites, when neutrally evolving, should approach near complete polymorphism saturation. If every neutral site is expected to carry at least one variant, the classic problem of distinguishing sites that are monomorphic due to chance (no mutation) versus sites that are monomorphic due to selective constraint (removed mutations) is greatly simplified. The point at which population genomic datasets are saturated with polymorphisms should represent a major advance in understanding the DFE at individual sites and is what immediately piqued my interest.

Overall, this manuscript is a thorough and thoughtful examination of this topic; to my enjoyment there were several times where a question came to mind that was addressed shortly later in the paper. I believe the authors have made a compelling case for why methylated CpG sites provide an entry point for understanding the site-specific DFE. I found the section "Interpreting monomorphic and polymorphic sites in current reference databases" particularly insightful as a guide to thinking about future datasets; similarly, I thought the comparison with CADD scores (Figure S9) provided important food for thought regarding confounders to maps of constraint generated from vast numbers of species using modern genomic datasets.

While the study is addressing an interesting topic, I also felt this manuscript was limited in novel findings to take away. Certainly the study clearly shows that substitution saturation is achieved at synonymous CpG sites. However, subsequent main analyses do not really show anything new: the depletion of segregating sites in functional versus neutral categories (Figure 2) has been extensively shown in the literature and polymorphism saturation is not a necessary condition for observing this pattern. Similarly, the diminishing returns on sampling new variable sites has been shown in previous studies, for example the first "large" human datasets ca. 2012 (e.g. Figure 2 in Nelson et al., 2012, Science) have similar depictions as Figure 3B although with smaller sample sizes and different approaches (projection vs simulation in this study). There are some simulations presented in Figure 4, but this is more of a hypothetical representation of the site-specific DFE under simulation conditions roughly approximating human demography than formal inference on single sites. Again, these all describe the state of the field quite well, but I was disappointed by the lack of a novel finding derived from exploiting the mutation saturation properties at methylated CpG sites.

Similarly, I felt the authors posed a very important point about limitations of DFE inference methods in the Introduction but ended up not really providing any new insights into this problem. The authors argue (rightly so) that currently available DFE estimates are limited by both the sparsity of polymorphisms and limited flexibility in parametric forms of the DFE. However, the nonsynonymous human DFE estimates in the literature appear to be surprisingly robust to sample size: older estimates (Eyre-Walker et al., 2006 Genetics, Boyko et al., 2008 PLOS Genetics) seem to at least be somewhat consistent with newer estimates (assuming the same mutation rate) from samples that are orders of magnitude larger (Kim et al., 2017 Genetics). Whether a DFE inferred under polymorphism saturation conditions with different methods is different, and how it is different, is an issue of broad and immediate relevance to all those conducting population genomic simulations involving purifying selection. The analyses presented as Figure 4A and 4B kind of show this, but they are more a demonstration of what information one might have at 1M+ sample sizes rather than an analysis of whether genome-wide nonsynonymous DFE estimates are accurate. In other words, this manuscript makes it clear that a problem exists, that it is a fundamental and important problem in population genetics, and that with modern datasets we are now poised to start addressing this problem with some types of sites, but all of this is already very well-appreciated except for perhaps the last point.

At least a crude analysis to directly compare the nonsynonymous genome-wide DFE from smaller samples to the 780K sample would be helpful, but it should be noted that these kinds of analyses could be well beyond the scope of the current manuscript. For example, if methylated nonsynonymous CpG sites are under a different level of constraint than other nonsynonymous sites (Figure S14) then comparing results to a genome-wide nonsynonymous DFE might not make sense and any new analysis would have to try and infer a DFE independently from synonymous/nonsynonymous methylated CpG sites.

Abstract: where it says "Here, we focus on putatively-neutral, synonymous CpG sites…" I thought the phrase "putatively-neutral, synonymous" could be clearer to the reader if moved to "… not seeing a polymorphism [at putatively-neutral, synonymous sites] is indicative of strong…".

Page 3 – "DNM" and "FET" were not defined before the first usage of the acronyms.

Page 7 – "That synonymous sites are close to saturation…": Here, wouldn't the expected length of the genealogy such that 1 mutation is expected per synonymous CpG site be a pretty drastic underestimate of the length of the genealogy such that saturation is observed (99% of synonymous CpG sites w/mutation)? Wouldn't a more precise estimate be something like 39 million generations, [1-Pois(0|1.17e-7*39e6)] ~ 99% of sites?

*Reviewer #2:*

This manuscript presents a simple and elegant argument that neutrally evolving CpG sites are now mutationally saturated, with each having a 99% probability of containing variation in modern datasets containing hundreds of thousands of exomes. The authors make a compelling argument that for CpG sites where mutations would create genic stop codons or impair DNA binding, about 20% of such mutations are strongly deleterious (likely impairing fitness by 5% or more). Although it is not especially novel to make such statements about the selective constraint acting on large classes of sites, the more novel aspect of this work is the strong site-by-site prediction it makes that most individual sites without variation in UK Biobank are likely to be under strong selection.

The authors rightly point out that since 99% of neutrally evolving CpG sites contain variation in the data they are looking at, a CpG site without variation is likely evolving under constraint with a p value significance of 0.01. However, a weakness of their argument is that they do not discuss the associated multiple testing problem-in other words, how likely is it that a given non synonymous CpG site is devoid of variation but actually not under strong selection? Since one of the most novel and useful deliverables of this paper is single-base-pair-resolution predictions about which sites are under selection, such a multiple testing correction would provide important "error bars" for evaluating how likely it is that an individual CpG site is actually constrained, not just the proportion of constrained sites within a particular functional category.

The paper provides a comparison of their functional predictions to CADD scores, an older machine-learning-based attempt at identifying site by site constraint at single base pair resolution. While this section is useful and informative, I would have liked to see a discussion of the degree to which the comparison might be circular due to CADD's reliance on information about which sites are and are not variable. I had trouble assessing this for myself given that CADD appears to have used genetic variation data available a few years ago, but obviously did not use the biobank scale datasets that were not available when that work was published.

Reading this paper left me excited about the possibility of examining individual invariant CpG sites and deducing how many of them are already associated with known disease phenotypes. I believe the paper does not mention how many of these invariant sites appear in Clinvar or in databases of patients with known developmental disorders, and I wondered how close to saturation disease gene databases might be given that individuals with developmental disorders are much more likely to have their exomes sequenced compared to healthy individuals. One could imagine some such analyses being relatively low hanging fruit that could strengthen the current paper, but the authors also make several reference to a companion paper in preparation that deals more directly with the problem of assessing clinical variant significance. This is a reasonable strategy, but it does give the Discussion section of the paper somewhat of a "to be continued" feel.

I think the paper could be strengthened by calculating the proportion of non-variable CpG sites in teach category are likely to be truly under constraint, making use of some kind of multiple testing correction. This would build upon the intuition that a non-variable CpG is likely functional with a non-corrected p value of 0.01.

My point about the possible circularity of comparison to CADD could be addressed with further discussion of the degree to which CADD is informed by patterns of human genetic variation and how incorporation of genetic variation into CADD scores might affect the conclusions of this section. As an additional point in the CADD section, it's not totally clear whether the statement "Mean transition rates at methylated CpGs are similar across CADD deciles" is based on de novo mutation data or some other data source.

Another addition that would add a lot to the paper, though is not strictly necessary, would be to comment on the overlap between sites identified as under selection by the current paper and sites where mutations are already annotated as clinically relevant or suspected to be so based on their occurrence in a disease cohort.

*Reviewer #3:*

Agarwal et al., combine a few well-known ideas in population genetics – diminishing returns in sampling new alleles with increasing sample size and the enrichment of invariant sites for sites under strong purifying selection – and point out the exciting result that sample sizes of modern human data sets are sufficiently large that, for highly mutable sites, saturation mutation has been reached. This is my favorite kind of result – one that is strikingly obvious in retrospect but that I had never considered (and probably wouldn't have). The manuscript is well written, and a number of my concerns or questions while reading were resolved directly by the authors later on. I have no major concerns, but a few potential suggestions that might strengthen the presentation.

The authors emphasize several times how important an accurate demographic model is. While we may be close to a solid demographic model for humans, this is certainly not the case for many other organisms. Yet we are not far off from sufficient sample sizes in a number of species to begin to reach saturation. I found myself wondering how different the results/inference would be under a different model of human demographic history. Though likely the results would be supplemental, it would be nice in the main text to be able to say something about whether results are qualitatively different under a somewhat different published model.

On a similar note, while a fixed hs simplifies much of the analysis, I wondered how results would differ for (1) completely recessive mutations and (2) under a distribution of dominance coefficients, especially one in which the most deleterious alleles were more recessive. Again, though I think it would strengthen the manuscript by no means do I feel this is a necessary addition, though some discussion of variation in dominance would be an easy and helpful add.

There's some discussion of population structure, but I also found myself wondering about GxE. That is, another reason a variant might be segregating is that it's conditionally neutral in some populations and only deleterious in a subset. I think no analysis to be done here, but perhaps some discussion?

Maybe I missed it, but I don't think the acronym DNM is explained anywhere. While it was fairly self-explanatory, I did have a moment of wondering whether it was methylation or mutation and can't hurt to be explicit.

---

## [Author Response]

All three reviewers were very enthusiastic about the manuscript, and with some revision it will clearly be suitable for publication in eLife.The main revision reviewers agreed would be important to incorporate is the issue of multiple testing (highlighted by Reviewer #2).While all the reviewers were enthusiastic about the manuscript as written, and none felt that additional analyses were necessary, there were a number of additional analyses suggested that reviewers felt could strengthen the paper if they were relatively straightforward to incorporate. These included:1) Comparison of the nonsynonymous genome-wide DFE from smaller samples to the 780K, perhaps only focusing on methylated nonsynomous sites.2) Comparison of overlap between these invariant sites and Clinvar or databases of patients with known developmental disorders.3) Comparison of results under a different parameterization of human demography.Reviewers had a number of additional suggestions that we felt would improve the manuscript. Perhaps first among these was a general feeling of "to be continued" in the discussion with multiple citations to another manuscript in prep. While reviewers were not against this strategy by any means, efforts to address the novelty of the current paper in the discussion (rather than simply hinting at exciting results in the forthcoming work) would be worthwhile.Please do review and respond to the individual reviewer comments as well.

We greatly appreciate the reviewers’ enthusiasm for the manuscript and thank them for their helpful comments.

We agree with the reviewers that it is important to address the question of how likely it is that a given non-synonymous mCpG site is invariant in current samples but nonetheless neutral. We had intended for this question to be addressed by our Bayesian analysis for an individual site in Figure 4, which examines how one’s beliefs about selection should be updated after observing a site to be invariant (i.e., using Bayes odds), with the p-value analogy serving simply to point out what makes mutation saturation special. We apologize for our failure to make this explicit, and have clarified the following points in the text:

A) First, we have now added a discussion of false discovery rates (FDR) in our mutation saturation based test for whether a non-syn site is neutral. As we now state, given that 1.2% of neutral sites are invariant, and 7.4% of all non-syn sites are invariant, the FDR across all non-synonymous invariant sites is ~16% (1.2/7.4). Within just LOFs it is ~4% (1.2/27).

B) With our demographic model and given our choice of prior, the Bayes odds for an invariant mCpG site at the current sample size are 15:1 in favor of hs > 0.5x10^-3^; thus there is a 1/16 chance an invariant site is not under “strong selection”.

We note that these odds depend on our prior and demographic model while the FDR calculation is based on the assumption that the null model is given by synonymous sites.

With regard to the three other points raised above, we have addressed (3) by showing results for the widely used model of Tennessen et al., (Tennessen et al., 2012), which was inferred from a relatively small sample size and thus did not detect the rapid exponential growth in population size towards the present; as expected, this model does not predict observed levels of variation as well and leads to different expectations about allele frequencies of deleterious alleles. In other words, and as expected, the results in Figure 4 depend on the demographic model choice.

We have also taken up suggestion (2) and now include an analysis of ClinVar and Deciphering Developmental Disorders (DDD) datasets. While both data sets are far from saturation, as expected, variants in those data sets, which are likely strongly deleterious, are enriched among invariant sites relative to segregating sites.

Regarding the analysis suggested in (1), we would argue we already know what will happen: given that small sample sizes carry very little information (see Figure 4a,b), the answer will depend greatly on the choice of parametric form for the DFE. That a priori choice is necessarily arbitrary, given that it is precisely what we are trying to learn about. Indeed, a recent preprint by Dukler et al., (Dukler et al., 2021) reports estimates of *sh* against amino-acid mutations inconsistent with those of Kim et al., (see p. 11 of their Discussion) and those of Kim et al., are in poor agreement with those of Boyko et al., and others (see Figure 4 in Kim et al., 2017).

Finally, we apologize for the impression that we postponed interesting analyses to a different paper, which is not a continuation of the current manuscript, but instead presents a complementary analysis of loss-of-function variation in human exomes and includes inference of a site-level DFE for such variants. We have now tried to make this clearer in the text, as well as adding a Figure on DDD and ClinVar and other analyses, as suggested.

Reviewer #1:Agarwal and Przeworski have performed a very timely and interesting study of the distribution of fitness effects (DFE) of new mutations. This study is timely because modern human population genetic datasets have finally achieved sample sizes for which a certain class of nucleotide sites, i.e. methylated CpG sites, when neutrally evolving, should approach near complete polymorphism saturation. If every neutral site is expected to carry at least one variant, the classic problem of distinguishing sites that are monomorphic due to chance (no mutation) versus sites that are monomorphic due to selective constraint (removed mutations) is greatly simplified. The point at which population genomic datasets are saturated with polymorphisms should represent a major advance in understanding the DFE at individual sites and is what immediately piqued my interest.Overall, this manuscript is a thorough and thoughtful examination of this topic; to my enjoyment there were several times where a question came to mind that was addressed shortly later in the paper. I believe the authors have made a compelling case for why methylated CpG sites provide an entry point for understanding the site-specific DFE. I found the section "Interpreting monomorphic and polymorphic sites in current reference databases" particularly insightful as a guide to thinking about future datasets; similarly, I thought the comparison with CADD scores (Figure S9) provided important food for thought regarding confounders to maps of constraint generated from vast numbers of species using modern genomic datasets.While the study is addressing an interesting topic, I also felt this manuscript was limited in novel findings to take away. Certainly the study clearly shows that substitution saturation is achieved at synonymous CpG sites. However, subsequent main analyses do not really show anything new: the depletion of segregating sites in functional versus neutral categories (Figure 2) has been extensively shown in the literature and polymorphism saturation is not a necessary condition for observing this pattern.

We agree with the reviewer that many of the points raised were appreciated previously and did not mean to convey another impression. Our aim was instead to highlight some unique opportunities provided by being at or very near saturation for mCpG transitions. In that regard, we note that although depletion of variation in functional categories is to be expected at any sample size, the selection strength that this depletion reflects is very different in samples that are far from saturated, where invariant sites span the entire spectrum from neutral to lethal. Consider the depletion per functional category relative to synonymous sites in Author response image 1 in a sample of 100k: ~40% of mCpG LOF sites do not have T mutations. From Figure 4, it can be seen that these sites are associated with a much broader range of *hs* values than sites invariant at 780K, so that information about selection at an individual site is quite limited (indeed, in our p-value formulation, these sites would be assigned p≤0.35, see Figure 1). Thus, only now can we really start to tease apart weakly deleterious mutations from strongly deleterious or even embryonic lethal mutations. This allows us to identify individual sites that are most likely to underlie pathogenic mutations and functional categories that harbor deleterious variation at the extreme end of the spectrum of possible selection coefficients. More generally, saturation is useful because it allows one to learn about selection with many fewer untested assumptions than previously feasible.

**Author response image 1. sa2fig1:** 

Similarly, the diminishing returns on sampling new variable sites has been shown in previous studies, for example the first "large" human datasets ca. 2012 (e.g. Figure 2 in Nelson et al., 2012, Science) have similar depictions as Figure 3B although with smaller sample sizes and different approaches (projection vs simulation in this study).

We agree completely: diminishing returns is expected on first principles from coalescent theory, which is why we cited a classic theory paper when making that point in the previous version of the manuscript. Nonetheless, the degree of saturation is an empirical question, since it depends on the unknown underlying demography of the recent past. In that regard, we note that Nelson et al., predict that at sample sizes of 400K chromosomes in Europeans, approximately 20% of all synonymous sites will be segregating at least one of three possible alleles, when the observed number is 29%. Regardless, not citing Nelson et al., 2012 was a clear oversight on our part, for which we apologize; we now cite it in that context and in mentioning the multiple merger coalescent.

There are some simulations presented in Figure 4, but this is more of a hypothetical representation of the site-specific DFE under simulation conditions roughly approximating human demography than formal inference on single sites. Again, these all describe the state of the field quite well, but I was disappointed by the lack of a novel finding derived from exploiting the mutation saturation properties at methylated CpG sites.

As noted above, in our view, the novelty of our results lies in their leveraging saturation in order to identify sites under extremely strong selection and make inferences about selection without the need to rely on strong, untested assumptions.

However, we note that Figure 4 is not simply a hypothetical representation, in that it shows the inferred DFE for single mCpG sites for a fixed mutation rate and given a plausible demographic model, given data summarized in terms of three ranges of allele frequency (i.e., = 0, between 1 and 10 copies, or above 10 copies). One could estimate a DFE across all sites from those summaries of the data (i.e., from the proportion of mCpG sites in each of the three frequency categories), by weighting the three densities in Figure 4 by those proportions. That is, in fact, what is done in a recent preprint by Dukler et al., (2021, BioRxiv): they infer the DFE from two summaries of the allele frequency spectrum (in bins of sites), the proportion of invariant sites and the proportion of alleles at 1-70 copies, in a sample of 70K chromosomes.

To illustrate how something similar could be done with Figure 4 based on individual sites, we obtain an estimate of the DFE for LOF mutations (shown in Author response image 2 Panel B and D for two different prior distributions on hs) by weighting the posterior densities in Panel A by the fraction of LOF mutations that are segregating (73% at 780K; 9% at 15K) and invariant (27% and 91% respectively); in panel C, we show the same for a different choice of prior. For the smaller sample size considered, the posterior distribution recapitulates the prior, because there is little information about selection in whether a site is observed to be segregating or invariant, and particularly about strong selection. In the sample of 780K, there is much more information about selection in a site being invariant and therefore, there is a shift towards stronger selection coefficients for LOF mutations regardless of the prior.

Our goal was to highlight these points rather than infer a DFE using these two summaries, which throw out much of the information in the data (i.e., the allele frequency differences among segregating sites). In that regard, we note that the DFE inference would be improved by using the allele frequency at each of 1.1 million individual mCpG sites in the exome. We outline this next step in the Discussion but believe it is beyond the scope of our paper, as it is a project in itself--in particular it would require careful attention to robustness with regard to both the demographic model (and its impact on multiple hits), biased gene conversion and variability in mutation rates among mCpG sites. We now make these points explicitly in the Outlook.

Similarly, I felt the authors posed a very important point about limitations of DFE inference methods in the Introduction but ended up not really providing any new insights into this problem. The authors argue (rightly so) that currently available DFE estimates are limited by both the sparsity of polymorphisms and limited flexibility in parametric forms of the DFE. However, the nonsynonymous human DFE estimates in the literature appear to be surprisingly robust to sample size: older estimates (Eyre-Walker et al., 2006 Genetics, Boyko et al., 2008 PLOS Genetics) seem to at least be somewhat consistent with newer estimates (assuming the same mutation rate) from samples that are orders of magnitude larger (Kim et al., 2017 Genetics).

We are not quite sure what the reviewer has in mind by “somewhat consistent,” as Boyko et al., estimate that 35% of non-synonymous mutations have s>10^-2^ while Kim et al., find that proportion to be “0.38–0.84 fold lower” than the Boyko et al., estimate (see, e.g., Figure 4 in Kim et al., 2017). Moreover, the preprint by Dukler et al., mentioned above, which infers the DFE based on ~70K chromosomes, finds estimates inconsistent with those of Kim et al., (see SOM Table 2 and SOM Figure S5 in Dukler et al., 2021).

More generally, given that even 70K chromosomes carry little information about much of the distribution of selection coefficients (see our Figure 4), we expect that studies based on relatively sample sizes will basically recover something close to their prior; therefore, they should agree when they use the same or similar parametric forms for the distribution of selection coefficients and disagree otherwise. The dependence on that choice is nicely illustrated in Kim et al., who consider different choices and then perform inference on the same data set and with the same fixed mutation rate for exomes; depending on their choice anywhere between 5%-28% of non-synonymous changes are inferred to be under strong selection with s>=10^-2^ (see their Table S4).

Whether a DFE inferred under polymorphism saturation conditions with different methods is different, and how it is different, is an issue of broad and immediate relevance to all those conducting population genomic simulations involving purifying selection. The analyses presented as Figure 4A and 4B kind of show this, but they are more a demonstration of what information one might have at 1M+ sample sizes rather than an analysis of whether genome-wide nonsynonymous DFE estimates are accurate. In other words, this manuscript makes it clear that a problem exists, that it is a fundamental and important problem in population genetics, and that with modern datasets we are now poised to start addressing this problem with some types of sites, but all of this is already very well-appreciated except for perhaps the last point.At least a crude analysis to directly compare the nonsynonymous genome-wide DFE from smaller samples to the 780K sample would be helpful, but it should be noted that these kinds of analyses could be well beyond the scope of the current manuscript. For example, if methylated nonsynonymous CpG sites are under a different level of constraint than other nonsynonymous sites (Figure S14) then comparing results to a genome-wide nonsynonymous DFE might not make sense and any new analysis would have to try and infer a DFE independently from synonymous/nonsynonymous methylated CpG sites.

We are not sure what would be learned from this comparison, given that Figure 4 shows that, at least with an uninformative prior, there is little information about the true DFE in samples, even of tens of thousands of individuals. Thus, if some of the genome-wide nonsynonymous DFE estimates based on small sample sizes turn out to be accurate, it will be because the guess about the parametric shape of the DFE was an inspired one. In our view, that is certainly possible but not likely, given that the shape of the DFE is precisely what the field has been aiming to learn and, we would argue, what we are now finally in a position to do for CpG mutations in humans.

Abstract: where it says "Here, we focus on putatively-neutral, synonymous CpG sites…" I thought the phrase "putatively-neutral, synonymous" could be clearer to the reader if moved to "… not seeing a polymorphism [at putatively-neutral, synonymous sites] is indicative of strong…".

We apologize for the unclear phrasing; we meant to say that given mutation saturation at putatively neutral sites, not seeing a *non-synonymous polymorphism* is indicative of strong selection against that mutation.We have now amended the text to make this clear.

Page 3 – "DNM" and "FET" were not defined before the first usage of the acronyms.

We have fixed this issue in the text.

Page 7 – "That synonymous sites are close to saturation…": Here, wouldn't the expected length of the genealogy such that 1 mutation is expected per synonymous CpG site be a pretty drastic underestimate of the length of the genealogy such that saturation is observed (99% of synonymous CpG sites w/mutation)? Wouldn't a more precise estimate be something like 39 million generations, [1-Pois(0|1.17e-7*39e6)] ~ 99% of sites?

We thank the reviewer for pointing out that we were being imprecise and have now updated the text to follow the suggestion.

Reviewer #2:This manuscript presents a simple and elegant argument that neutrally evolving CpG sites are now mutationally saturated, with each having a 99% probability of containing variation in modern datasets containing hundreds of thousands of exomes. The authors make a compelling argument that for CpG sites where mutations would create genic stop codons or impair DNA binding, about 20% of such mutations are strongly deleterious (likely impairing fitness by 5% or more). Although it is not especially novel to make such statements about the selective constraint acting on large classes of sites, the more novel aspect of this work is the strong site-by-site prediction it makes that most individual sites without variation in UK Biobank are likely to be under strong selection.The authors rightly point out that since 99% of neutrally evolving CpG sites contain variation in the data they are looking at, a CpG site without variation is likely evolving under constraint with a p value significance of 0.01. However, a weakness of their argument is that they do not discuss the associated multiple testing problem-in other words, how likely is it that a given non synonymous CpG site is devoid of variation but actually not under strong selection? Since one of the most novel and useful deliverables of this paper is single-base-pair-resolution predictions about which sites are under selection, such a multiple testing correction would provide important "error bars" for evaluating how likely it is that an individual CpG site is actually constrained, not just the proportion of constrained sites within a particular functional category.

We thank the reviewer for pointing this out. As we outline in response to the editorial comments, one way to think about this problem might be in terms of false discovery rates, in which case the FDR would be 16% across all non-synonymous mCpG sites that are invariant in current samples, and ~4% for the subset of those sites where mutations lead to loss-of-function of genes.

Another way to address this issue, which we had included but not emphasized previously, is by examining how one’s beliefs about selection should be updated after observing a site to be invariant (i.e., using Bayes odds). At current sample sizes and assuming our uninformative prior, for a non-synonymous mCpG site that does not have a C>T mutation, the Bayes odds are 15:1 in favor of *hs*>0.5x10^-3^; thus the chance that such a site is not under strong selection is 1/16, given our prior and demographic model.

As noted in response to the editor, these two approaches (FDR and Bayes odds) are based on somewhat distinct assumptions.

We have now added and/or emphasized these two points in the main text.

The paper provides a comparison of their functional predictions to CADD scores, an older machine-learning-based attempt at identifying site by site constraint at single base pair resolution. While this section is useful and informative, I would have liked to see a discussion of the degree to which the comparison might be circular due to CADD's reliance on information about which sites are and are not variable. I had trouble assessing this for myself given that CADD appears to have used genetic variation data available a few years ago, but obviously did not use the biobank scale datasets that were not available when that work was published.

We apologize for the lack of clarity in the presentation. We meant to emphasize that de novo *mutation rates* vary across CADD deciles when considering all CpG sites (Figure 2—figure supplement 5c), which confounds CADD precisely because it is based in part on which sites are variable. We now write:

“We can also check that the fraction of sites segregating is inversely proportional to the predicted functional importance of the sites using CADD scores (12), widely used measures of constraint that incorporate functional annotations and measures of conservation. Across deciles, mean de novo transition rates at methylated CpGs are similar (Figure 2—figure supplement 5a) and, as expected, the fraction of segregating sites decreases with increasing CADD scores (Figure 2—figure supplement 5b). We note, however, that mutation rates may not always be similar across comparison groups: considering all CpG sites in exons (i.e., not only highly methylated ones), for example, de novo mutation rates are much more variable across CADD deciles (Figure 2—figure supplement 5c). Consequently the depletion of segregating sites no longer has a simple interpretation (Figure 2—figure supplement 5d), instead reflecting a combination of differences in mutation rates and fitness effects. By implication, while CADD scores are meant to isolate the effects of selection, they will in some cases classify sites that have high mutation rates as less constrained, and vice versa.”

Reading this paper left me excited about the possibility of examining individual invariant CpG sites and deducing how many of them are already associated with known disease phenotypes. I believe the paper does not mention how many of these invariant sites appear in Clinvar or in databases of patients with known developmental disorders, and I wondered how close to saturation disease gene databases might be given that individuals with developmental disorders are much more likely to have their exomes sequenced compared to healthy individuals. One could imagine some such analyses being relatively low hanging fruit that could strengthen the current paper, but the authors also make several reference to a companion paper in preparation that deals more directly with the problem of assessing clinical variant significance. This is a reasonable strategy, but it does give the Discussion section of the paper somewhat of a "to be continued" feel.

We apologize for the confusion that arose from our references to a second manuscript in prep. The companion paper is not a continuation of the current manuscript: it contains an analysis of fitness and pathogenic effects of loss-of-function variation in human exomes.

Following the reviewer’s suggestion to address the clinical significance of our results, we have now examined the relationship of mCpG sites invariant in current samples with ClinVar variants. We find that of the approximately 59,000 non-synonymous mCpG sites that are invariant, only ~3.6% overlap with C>T variants associated with at least one disease and classified as likely pathogenic in ClinVar (~5.8% if we include those classified as uncertain or with conflicting evidence as pathogenic). Approximately 2% of invariant mCpGs have C>T mutations in what is, to our knowledge, the largest collection of de novo variants ascertained in ~35,000 individuals with developmental disorders (DDD, Kaplanis et al., 2020). At the level of genes, of the 10k genes that have at least one invariant non-synonymous mCpG, only 8% (11% including uncertain variants) have any non-synonymous hits in ClinVar, and ~8% in DDD. We think it highly unlikely that the large number of remaining invariant sites are not seen with mutations in these databases because such mutations are lethal; rather it seems to us to be the case that these disease databases are far from saturation as they contain variants from a relatively small number of individuals, are subject to various ascertainment biases both at the variant level and at the individual level, and only contain data for a small subset of existing severe diseases.

With a view to assessing clinical relevance however, we can ask a related question, namely how informative being invariant in a sample of 780K is about pathogenicity in ClinVar. Although the relationship between selection and pathogenicity is far from straightforward, being an invariant non-synonymous mCpG in current samples not only substantially increases (15-fold) the odds of hs > 0.5x10^-3^ (see Figure 4b), it also increases the odds of being classified as pathogenic vs. benign in ClinVar 8-51 fold. In the DDD sample, we don’t know which variants are pathogenic; however, if we consider non-synonymous mutations that occur in consensus DDD genes as pathogenic (a standard diagnostic criterion), being invariant increases the odds of being classified as pathogenic 6-fold. We caution that both ClinVar classifications and the identification of consensus genes in DDD relies in part on whether a site is segregating in datasets like ExAC, so this exercise is somewhat circular. Nonetheless it illustrates that there is some information about clinical importance in mCpG sites that are invariant in current samples, and that the degree of enrichment (6 to 51-fold) is very roughly on par with the Bayes odds that we estimate of strong selection conditional on a site being invariant. We have added these findings to the main text and added the plot as Figure 4—figure supplement 2.

I think the paper could be strengthened by calculating the proportion of non-variable CpG sites in teach category are likely to be truly under constraint, making use of some kind of multiple testing correction. This would build upon the intuition that a non-variable CpG is likely functional with a non-corrected p value of 0.01.

As we write above, we now address this concern in the paper in two ways: using false discovery rates and Bayes odds of *hs*>0.5x10^-3^.

My point about the possible circularity of comparison to CADD could be addressed with further discussion of the degree to which CADD is informed by patterns of human genetic variation and how incorporation of genetic variation into CADD scores might affect the conclusions of this section. As an additional point in the CADD section, it's not totally clear whether the statement "Mean transition rates at methylated CpGs are similar across CADD deciles" is based on de novo mutation data or some other data source.

We apologize for the lack of clarity in the presentation but meant to emphasize that de novo mutation rates vary across all CpG sites by CADD score, which confounds CADD precisely because it is based in part on which sites are variable. We have now revised the text to hopefully clarify this point (see response to reviewer 1).

Another addition that would add a lot to the paper, though is not strictly necessary, would be to comment on the overlap between sites identified as under selection by the current paper and sites where mutations are already annotated as clinically relevant or suspected to be so based on their occurrence in a disease cohort.

We thank the reviewer for this suggestion. As detailed in our response above, we now show that CpG sites invariant in population cohorts that overlap with ClinVar and among de novo variants in the DDD cohort are informative about pathogenicity of variants. We have added Figure 4—figure supplement 2 to illustrate this point.

Reviewer #3:Agarwal et al., combine a few well-known ideas in population genetics – diminishing returns in sampling new alleles with increasing sample size and the enrichment of invariant sites for sites under strong purifying selection – and point out the exciting result that sample sizes of modern human data sets are sufficiently large that, for highly mutable sites, saturation mutation has been reached. This is my favorite kind of result – one that is strikingly obvious in retrospect but that I had never considered (and probably wouldn't have). The manuscript is well written, and a number of my concerns or questions while reading were resolved directly by the authors later on. I have no major concerns, but a few potential suggestions that might strengthen the presentation.The authors emphasize several times how important an accurate demographic model is. While we may be close to a solid demographic model for humans, this is certainly not the case for many other organisms. Yet we are not far off from sufficient sample sizes in a number of species to begin to reach saturation. I found myself wondering how different the results/inference would be under a different model of human demographic history. Though likely the results would be supplemental, it would be nice in the main text to be able to say something about whether results are qualitatively different under a somewhat different published model.

We had previously examined the effect of a few demographic scenarios with large increases in population size towards the present on the average length of the genealogy of a sample (and hence the expected number of mutations at a site) in Figure 3—figure supplement 1b, but without quantifying the effect on our selection inference. Following this suggestion, we now consider a widely used model of human demography inferred from a relatively small sample, and therefore not powered to detect the huge increase in population size towards the present (Tennessen et al., 2012). Using this model, we find a poor fit to the proportion of segregating CpG sites (the observed fraction is 99% in 780K exomes, when the model predicts 49%). Also, as expected, inferences about selection depend on the accuracy of the demographic model (as can be seen by comparing Author response image 3 panel B to Figure 4B in the main text).

**Author response image 3. sa2fig3:** 

On a similar note, while a fixed hs simplifies much of the analysis, I wondered how results would differ for (1) completely recessive mutations and (2) under a distribution of dominance coefficients, especially one in which the most deleterious alleles were more recessive. Again, though I think it would strengthen the manuscript by no means do I feel this is a necessary addition, though some discussion of variation in dominance would be an easy and helpful add.There's some discussion of population structure, but I also found myself wondering about GxE. That is, another reason a variant might be segregating is that it's conditionally neutral in some populations and only deleterious in a subset. I think no analysis to be done here, but perhaps some discussion?

We agree that our analysis ignores the possibilities of complete recessivity in fitness (*h*=0) as well as more complicated selection scenarios, such as spatially-varying selection (of the type that might be induced by GxE). We note however that so long as there are any fitness effects in heterozygotes, the allele dynamics will be primarily governed by *hs;* one might also imagine that under some conditions, the mean selection effect across environments would predict allele dynamics reasonably well even in the presence of GxE. Also worth exploring in our view is the standard assumption that *hs* remains fixed even as *N*_e_ changes dramatically. We now mention these points in the Outlook.

Maybe I missed it, but I don't think the acronym DNM is explained anywhere. While it was fairly self-explanatory, I did have a moment of wondering whether it was methylation or mutation and can't hurt to be explicit.

We apologize for the oversight and have updated the text accordingly.